

# Design of a pollution ontology-based event generation framework for the dynamic application of traffic restrictions

David Eneko Ruiz de Gauna[1], Luís Enrique Sánchez[2] and Almudena Ruiz-Iniesta[1]

[1] International University of La Rioja, Logroño, La Rioja, Spain
[2] University of Castilla-La Mancha, Ciudad Real, Castilla-La Mancha, Spain

## ABSTRACT

The environmental damage caused by air pollution has recently become the focus of city council policies. The concept of the green city has emerged as an urban solution by which to confront environmental challenges worldwide and is founded on air pollution levels that have increased meaningfully as a result of traffic in urban areas. Local governments are attempting to meet environmental challenges by developing public traffic policies such as air pollution protocols. However, several problems must still be solved, such as the need to link smart cars to these pollution protocols in order to find more optimal routes. We have, therefore, attempted to address this problem by conducting a study of local policies in the city of Madrid (Spain) with the aim of determining the importance of the vehicle routing problem (VRP), and the need to optimise a set of routes for a fleet. The results of this study have allowed us to propose a framework with which to dynamically implement traffic constraints. This framework consists of three main layers: the data layer, the prediction layer and the event generation layer. With regard to the data layer, a dataset has been generated from traffic data concerning the city of Madrid, and deep learning techniques have then been applied to this data. The results obtained show that there are interdependencies between several factors, such as weather conditions, air quality and the local event calendar, which have an impact on drivers' behaviour. These interdependencies have allowed the development of an ontological model, together with an event generation system that can anticipate changes and dynamically restructure traffic restrictions in order to obtain a more efficient traffic system. This system has been validated using real data from the city of Madrid.

# INTRODUCTION

## Motivation

Urban traffic management is a considerable socio-economic challenge that has a direct impact on cities and metropolitan areas. The impact of traffic congestion is a negative factor because it increases pollution levels and travel time (*Kong et al., 2016*). Moreover, solving transportation problems will help to reduce traffic jam levels, pollution levels,

Corresponding author
David Eneko Ruiz de Gauna,
davideneko.ruiz@unir.net

investments in infrastructures, and the maintenance of road and transportation systems (*Kuang et al., 2019*). Studying these decision-making issues is an important branch of research on urban intelligence that is required in order to build a model-based approach (*An, Jennings & Li, 2017*). The applications currently available combine the usefulness of real time systems that inform the user of certain conditions in urban areas (*Silva, Analide & Novais, 2014*) and 5G networks with which to improve network capacity in network congestion scenarios (*Cheng et al., 2018*). Open Data has, in recent years, become a transparent and frequently-adopted tool that allows citizens to share historical and real-time data regarding cities. This tool is aligned with an elite group of intelligent urban-planning pioneers that are leading the transformation of cities such as Singapore, Vienna, Barcelona, and Tokyo into more democratic and sophisticated metropolitan areas (*von Richthofen, Tomarchio & Costa, 2019*). For instance, in Spain, the Madrid City Council provides information related to different fields such as traffic flow, weather conditions, air pollution levels or noise levels, among others (*Madrid City Council, 2022*). This information has already been used to predict air quality levels depending on traffic flow conditions (*Lana et al., 2016*). The air pollution predictions in London are linked with traffic conditions (*Maciag et al., 2019*). In this respect, *Shams et al. (2021)* correlated the influence of traffic flow with $NO_2$, and in Wroclaw, the temporal correlation between meteorological and air pollution conditions is considered on the basis of traffic flow (*Brunello et al., 2019*). Many cities such as Barcelona, Madrid or Vienna are accordingly extending traffic restriction policies so as to improve their air quality. The publication of scientific articles addressing this issue is very recent, but some research works have provided interesting contributions and limitations. For instance, *Borge et al. (2018)* measured the impact of the Madrid $NO_2$ protocol using a multi-scale air quality model and a street-level approach for one of the main streets in the city. The results obtained after using different estimated traffic demand scenarios showed that only the most restrictive measure would produce a noticeable improvement in air quality. Moreover, this study had some limitations since it did not consider the local calendar holidays, along with the need to implement dynamic and permanent measures. Aligned with the limitations of static traffic models, *Tsanakas, Ekström & Olstam (2020)* designed a methodology with which to overcome the tendency of traffic conditions to underestimate emissions aggregated over time and space. They therefore considered that it would, in the future, be interesting to estimate emissions dynamically in order to load and evaluate the effect for applications such as air quality modelling. The first approach to consider the implementation of local policies was introduced in *Rodriguez Rey et al. (2021)*. These authors presented a modelling system based on the city-wide Low Emission Zone (LEZ), which restricts the entry of the most polluting vehicles into Barcelona. They evidenced the insufficiency of these local policies as regards complying with EU air quality standards owing to their lack of dynamism. Moreover, they presented some limitations in order to discover the impact of additional measures that are not being considered by Barcelona city council, such as the implementation of a congestion charge.

Previous works have presented interesting approaches related to traffic and pollution. Most of them have provided different scenarios in order to demonstrate the importance of

designing traffic restrictions with which to reduce pollution levels. Furthermore, they have provided limitations in order to apply dynamic scenarios for smart cars and implement specific variables such as congestion charges or local calendar holidays. There is consequently still a need to design and validate dynamic traffic restrictions on the basis of future pollution levels associated with traffic status so as to optimise a set of routes for a fleet.

We therefore present this work in order to address these open issues. Our work is organised in the following sections: the Related Works section, which provides a summary of the work being carried out in this area; the Methods section, which explains the methodology employed; the Results section, which shows dataset information and a performance evaluation; the Discussion section, in which the significance of our findings is described and interpreted, and finally, the Conclusion section, which presents the strengths and limitations of our work, along with promising directions for future work.

## Contribution

In an attempt to overcome the aforementioned limitations, we present an ontology based on the Madrid City Pollution Protocol, whose purpose is to facilitate the design of a methodology for the dynamic adoption of traffic restrictions. This approach is based on the connectionist learning and inter-ontology similarities (CILIOS) (*Rosaci, 2007*), who suggested the integration of an ontology in order to represent concepts, functions and causal implications among events in a multi-agent environment by using a mechanism that is capable of inducing logical rules representing agent behaviour on the basis of neural networks. Neural networks are frequently integrated into forecasting systems because the information is transformed into a sinusoidal time-varying signal and can have a proper working frequency (*Pappalardo et al., 1998*). The use of these fundamentals as a basis is, to the best of our knowledge, the first attempt to combine local mobility policies with pollution levels by employing an ontology based on data generated by neural networks. In this research, we used historical open data obtained from Madrid city council, thus allowing us to provide our ontology with real data. The last full year with all the data for every month is 2022, and this year has, therefore been used as a reference in our work. These real data have been used to provide a detailed statistical analysis of the relationship between air pollution, atmospheric variables, the local calendar and road traffic status. LSTM-RNN for time-series pollution level forecasting was also applied in this work by benchmarking the most frequently used forecasting algorithms. This choice was based on the best mean absolute error (MAE) between the validation and the test. Finally, we provided a dashboard, which was used to test our ontology with historical and future dates so as to validate our proposal.

## RELATED WORK

This section provides a summary of the existing work on traffic forecasting. The majority of the work uses the long short-term memory (LSTM) recurrent neural network (RNN). For example, *Kang, Lv & Chen (2018)* employ various input settings in LSTM-RNN predictions, which include traffic flow, speed and occupancy. In their work, *Bogaerts et al.*

*(2020)* use graph convolution to provide the spatial features of traffic and its temporal features by means of long short term memory (LSTM). In addition, some authors include weather conditions as an important feature, such as *Awan, Minerva & Crespi (2020a)*, who propose an LSTM-RNN with which to improve road traffic forecasting using air pollution and atmospheric data in Madrid. In Tehran (Iran), *Sadeghi-Niaraki et al. (2020)* made a classification of weather conditions, weekdays, and hour days and presented a short-term flow traffic prediction model based on the modified Elman recurrent neural network model (GA-MENN). Some authors have used other approaches. For example, *Zhou et al. (2021)* combined RNNs and convolutional neural networks (CNNs) by sliding windows over a map, while *Vélez-Serrano et al. (2021)* proposed CNNs for the spatio-temporal structure of a set of sensors. In their work, *Zhang & Kabuka (2018)* used RNN to predict traffic flow on the basis of weather conditions such as temperature, smoke or wind speed. In *Hou, Deng & Cui (2021)*, the temporal correlation and periodicity of traffic flow data were integrated with the disturbance of weather factors in order to compare different deep learning algorithms, while *Essien et al. (2021)* proposed a new approach in which social networks such as Twitter were used to predict the traffic flow in Greater Manchester, United Kingdom. The work of *Brunello et al. (2019)* shows how weather and air quality conditions were correlated in order to predict traffic flow in Wroclaw, Poland, and *Lee, Jeon & Sohn (2021)* used spatial-temporal correlations to develop a traffic-speed predictive methodology in Gangnam, Seoul, Korea. According to *Ma, Song & Li (2021)*, some of the factors in inter- and intra-day traffic patterns can be critical. In Guiyang, Guizhou province, China, *Wang et al. (2020)* implemented a traffic flow prediction method by combining the Group method of data handling (GMDH) with the seasonal autoregressive integrated moving average (SARIMA). In their work, *Bie et al. (2017)* used loop detectors to collect data on Whitemud Drive, Edmonton, Canada, and introduced weather factors in order to free flow speed, capacity, and critical density in the METANET model, while *Shahid et al. (2021)* proposed a framework in which regression models were used to forecast air pollution caused by road traffic. However, most works have not considered pollution as key feature for their predictions. To address this issue, few works have applied ontologies to improve smart cities environments (*Espinoza-Arias, Poveda-Villalón & Corcho, 2020*; *Mulero Martínez et al., 2018*). For instance, *Gonzalez-Mendoza, Velasco-Bermeo & López Orozco (2018)* represented a pollution and traffic knowledge within a domain through ontologies. These ontologies classify things, actions, features based on traffic status and pollution status. This approach is very useful to implement control traffic systems in smart cities to reduce pollution levels. Accordingly, many cities have recently published public protocols with which to combat traffic pollution levels. We believe that pollution levels affect traffic patterns because these protocols activate restrictions related to speed, access to districts and parking. The pollution protocol model has, to the best of our knowledge, not yet been studied in literature. We believe that it is necessary to understand the importance of protocols in order to improve traffic forecasting.
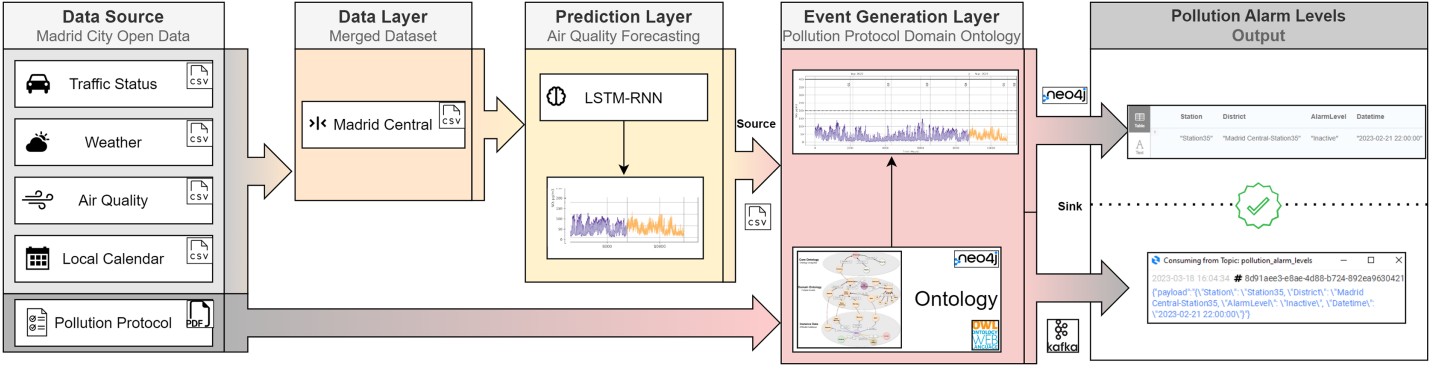

**Figure 1 Pollution ontology-based event generation framework.**

## METHODS

The objective of this article is to design a methodology for the dynamic adoption of traffic restrictions. As shown in Fig. 1, our framework consists of three layers: Data Layer, Prediction Layer, and Event Generation Layer. A more detailed explanation of our framework is provided as follows. The Data Layer extracts all the data from the Madrid City Open Data portal in csv format, and these files are consumed by the Prediction Layer. This layer ingests this data and its output is a forecast. This forecast, which is generated by a time series ML algorithm, returns future pollution values. These predictions are inferred by the ontology in the Event Generation Layer. This ontology internally normalises the official pollution protocol published by the Madrid City Council in 2020. This protocol establishes different alarm levels depending on the air quality values: In the range of [180–200) mg/$m^3$ are Early Warning level, in the range of [200–400) mg/$m^3$ are Warning level, and values greater than 400 mg/$m^3$ are Alert level. Therefore, the output of this ontology is an alarm pollution level (Inactive, Early Warning, Warning, and Alert) according to this pollution protocol.

### Data layer

This layer collects historical data in order to feed our framework. First, a large set of data regarding traffic intensities and pollution levels was stored, after which we collected data from the Madrid City Council's Open Data website (*Madrid City Council, 2022*) in CSV format. Moreover, we extracted data related to weather conditions and local calendar from the same source. This raw data is provided by more than 4,000 traffic intensity sensors, 26 weather stations and 24 air quality stations. For our prediction case, we focused on Madrid Central. This area constitutes the urban core, which consists of around 50 traffic intensity sensors together with one station for weather and air quality (Fig. 2).

Traffic status data in Madrid covers the entire metropolitan area of the city. As shown in Table 1, we filtered the data of those stations that are inside the target area. From an initial number of 4,000 traffic sensors, around 50 have been chosen that are located within Central Madrid. We extracted the main values that are related to traffic from these stations:
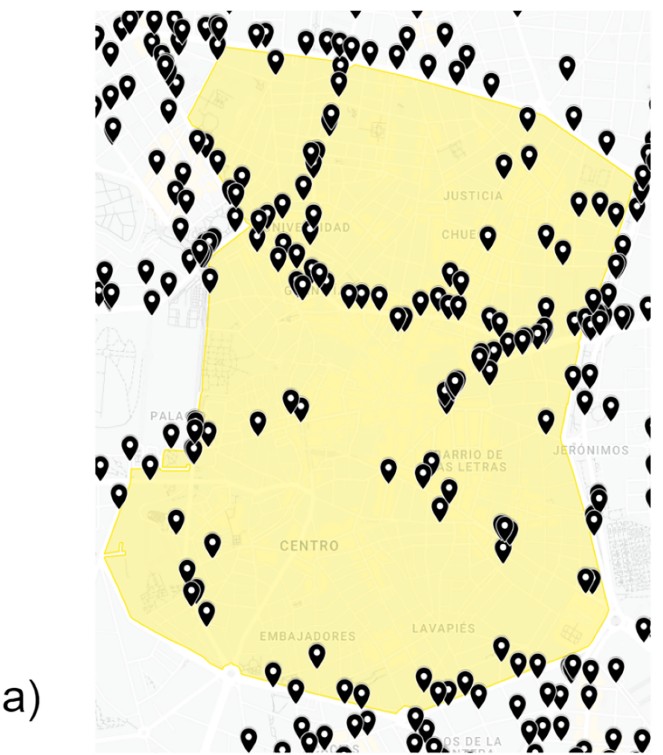
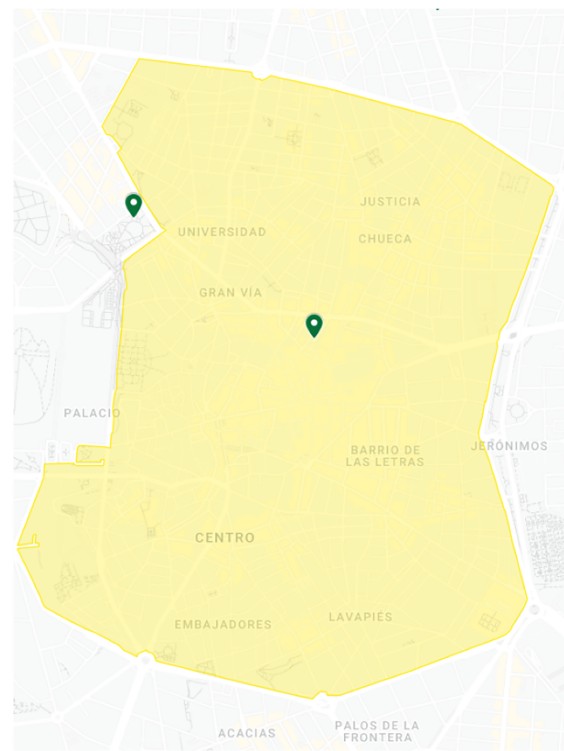

**Figure 2 Traffic intensity sensors (A), weather-air quality stations (B) in Madrid Central (yellow color).** Google, Instituto Geográfico Nacional (*Google, 2022*).

**Table 1 Dataset analysis of rows and columns.**

| Dataset name | No. rows (Original) | No. rows (Filtered) | No. columns (Original) | No. columns (Filtered) |
|---|---|---|---|---|
| Traffic status | 28,247,866 | 8,760 | 13 | 6 |
| Weather | 33,852 | 8,760 | 12 | 3 |
| Air quality | 1,332,889 | 8,760 | 13 | 3 |
| Local calendar | 28,567 | 365 | 19 | 3 |
| Merged dataset (prediction input) | 8,760 | 8,760 | 9 | 10* |

**Note:**
\* These columns are explained in the Prediction Layer section (see Table 2).

intensity and occupancy. Intensity is the number of vehicles that pass through a fixed section in a given time, while occupancy is the ratio between vehicles and the distance between vehicles on the road per lane, represented as a percentage. Accordingly, we reduced the number of columns compared to the original. In this case, we reduced the ones that we considered important from 13 columns to six. The selected columns are the following:

- *id*: Sensor identification code (integer)
- *date_timestamp*: Date timestamp (timestamp)

- *date_id*: Date time (integer)
- *hour_of_day*: Hour time (integer)
- *intensity*: Traffic intensity value (double)
- *occupancy*: Traffic occupancy value (double)

Currently, there are 26 weather stations in Madrid, but we filtered all the data to collect the data referring to the Madrid Central district. Afterwards, we merged temperature and humidity values due to their importance in our study. The final dataset have been reduced from initial 12 columns to three.

- *date_id*: Date time (integer)
- *humidity*: Weather humidity value (double)
- *temperature*: Weather temperature value (double)

Air quality data represents the urban levels detected by 24 stations. In the same way as with the weather data, we used the data related to the Madrid Central district. The original data comes with several magnitudes related to air quality, but we focused on the magnitude that is established in the pollution protocol. The Madrid pollution protocol uses only the $NO_2$ magnitude to apply different restrictions and this motivates the use of this magnitude in our study. As in the previous cases, we decreased the number of columns from 13 to three:

- *date_id*: Date time (integer)
- *hour*: Hour time (integer)
- *value*: $NO_2$ value (Double)

Local calendar data includes all public holidays and working days throughout the year in Madrid. This distinction is important to be able to distinguish how labor mobility influences traffic. We selected three columns out of 19 possible columns:

- *date_id*: Date time (integer)
- *day_of_week*: Day of the week (integer)
- *is_holiday*: Working/Holiday (integer)

Merged dataset combines the information needed by our architecture. This match is based on date and time. We used *date_id* to make this combination possible and build this dataset. The columns are as follows:

- *date_timestamp*: Date timestamp (timestamp)
- *hour_of_day*: Hour time (integer)
- *day_of_week*: Day of the week (integer)
- *is_holiday*: Working/Holiday (integer)
- *no₂_level*: $NO_2$ value (Double)
- *humidity*: Weather humidity value (double)

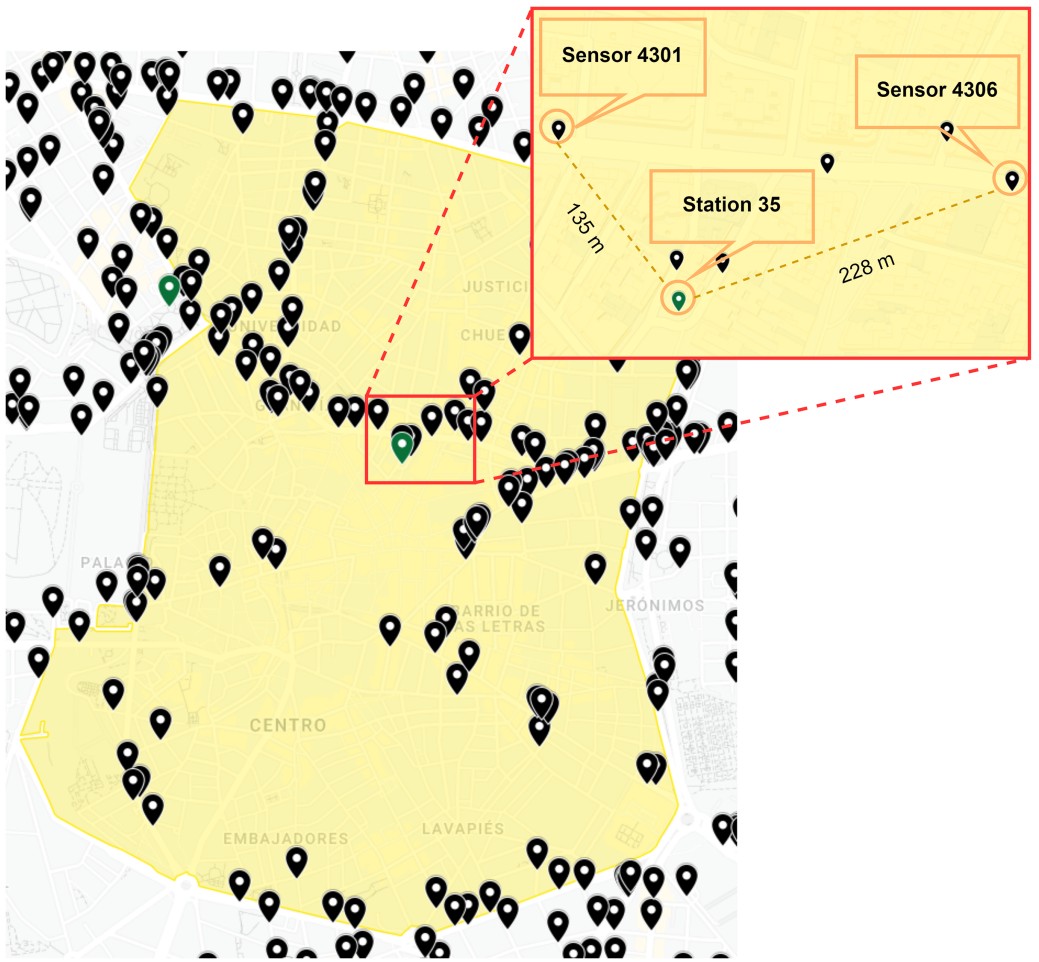

**Figure 3 Location of traffic intensity sensors and weather-air quality stations selected.** Google, Instituto Geográfico Nacional (*Google, 2022*).               

- *temperature*: Weather temperature value (double)
- *intensity*: Traffic intensity value (double)
- *occupancy*: Traffic occupancy value (double)

## Prediction layer

This layer represents our pollution level forecasting. Our analysis is divided into two categories: (1) statistical-based feature selection analysis and (2) predictive analysis. We first correlated weather, local calendar and air quality conditions with traffic intensity status. This approach makes it possible to explain the choice of features and their influence on traffic. We then collected and analysed metrics in order to attain a highly accurate pollution level using LSTM-RNN.

### Statistical-based feature selection analysis

As illustrated in Fig. 3, we randomly selected two traffic intensity sensors (4301 and 4306) in a 250–m perimeter of the weather and air quality station (35) in order to explain the
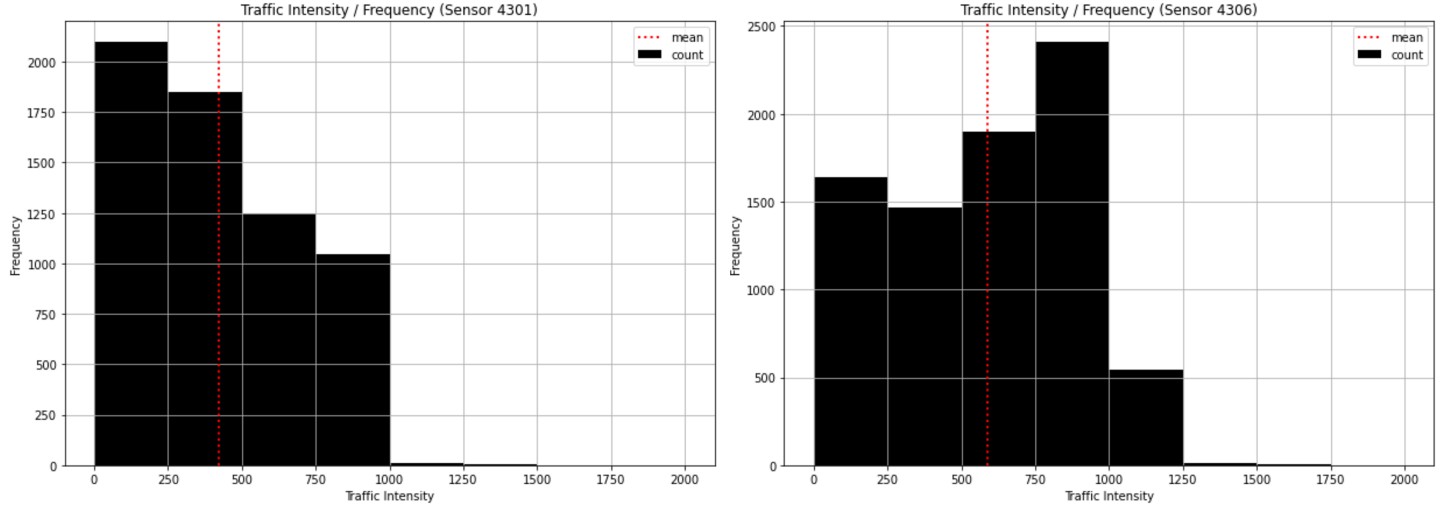

**Figure 4 Traffic intensity histogram for sensor 4301 and sensor 4306.**

correlation of the different features. This selection depended on the analysis of the aggregated data in order to understand any possible correlations among weather, air quality and the local calendar, and the hourly traffic status obtained from nearby traffic sensors. These correlations were then accordingly plotted in order to visualise patterns.

Our first analysis consisted of examining the most frequent traffic intensities. As illustrated in Fig. 4, the most frequent traffic intensity at Sensor 4301 is between 0 and 250, which appears more than 2,000 times. The least frequent is between 1,250 and 1,500, less than 100 times, while the mean is slightly higher than 400 vehicles/hour. This sensor, therefore steadily decreases from 250 to 1,000, but there are some low values above 1,000 that appear infrequently. In the case of sensor 4306, the most frequent is between 750 and 1,000, which occurs more than 2,000 times. The least frequent is between 1,500 and 1,750, less than 100 times, while the mean is slightly lower than 600 vehicles/hour. This sensor does not behave in a regular manner and consequently has a significant effect on the results. This signifies that some other variables are influencing traffic intensity.

Our second analysis focused on weather conditions. Figure 5 shows the correlation between temperature and traffic intensity. These values are the mean values obtained for both sensors. At first glance, the lower the temperature, the higher the number of vehicles passing by sensor 4301. Even if in the case of the 4306 sensor, the ranges (25–30] and (30–35) also present higher values. These optimal values are located roughly between the ranges (5–10] and (10–15]. With regard to sensor 4301, the highest value is located within the range (10–15], while the lowest values are influenced by temperatures of over 20 degrees. Sensor 4306 behaves in a similar manner to sensor 4301, but the lowest values are within the range (15–20]. The temperature consequently affects urban traffic intensity, thus it is necessary to study other variables that, together with temperature, can better explain certain differences in other results between the two sensors.

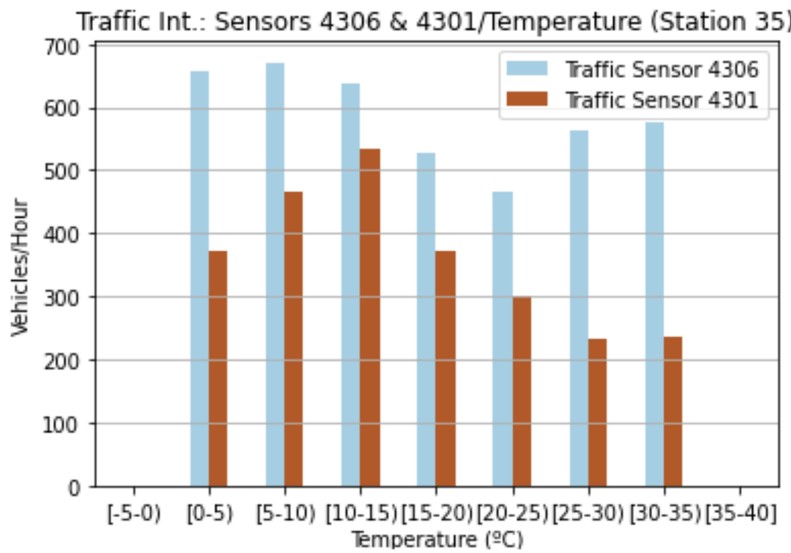

**Figure 5 Traffic intensity and temperature for sensor 4301 and sensor 4306.**

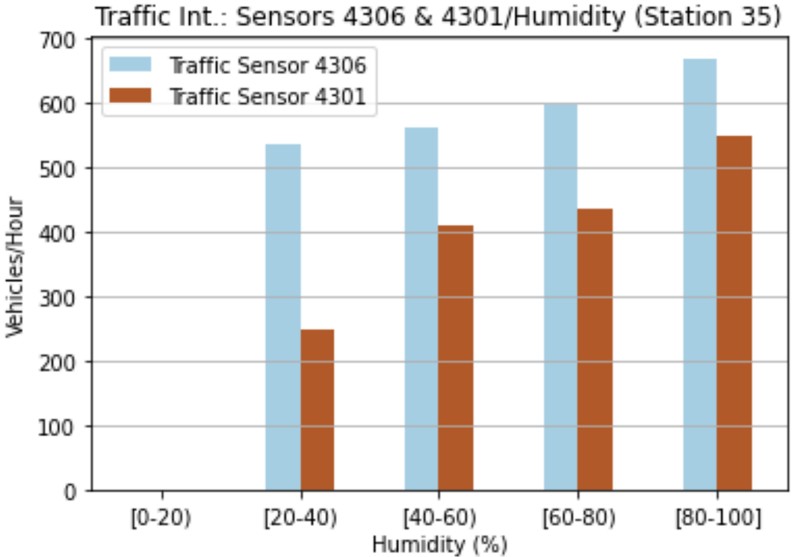

**Figure 6 Traffic intensity and humidity for sensor 4301 and sensor 4306.**

The third analysis was based on humidity in order to estimate rainfall days. As shown in Fig. 6, the humidity affects the traffic intensity at sensors 4301 and 4306. The higher the humidity, the higher the number of vehicles detected by the sensor. On the one hand, the highest values are located in the range [80–100) for both sensors, while on the other, the lowest values are in the range [20–40]. These results evidence the importance of humidity and temperatures as regards predicting future traffic intensities.

In our fourth analysis, we analysed the importance of local calendar traffic intensities. The bank holidays in Spain can fall on any day of the week and there may also be different

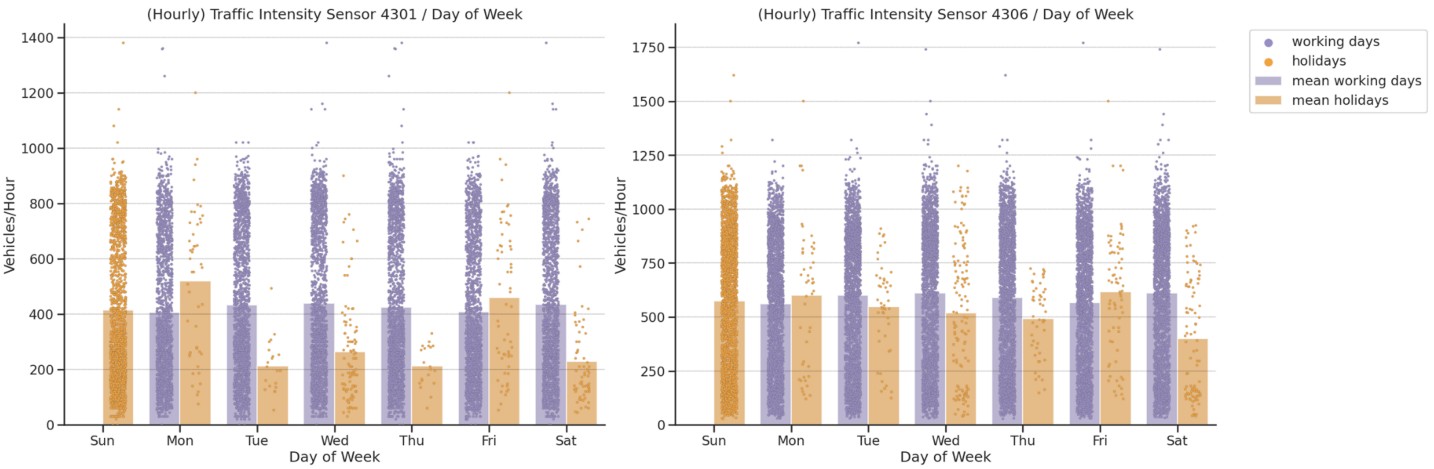

**Figure 7 Traffic intensity correlation by day of week.**

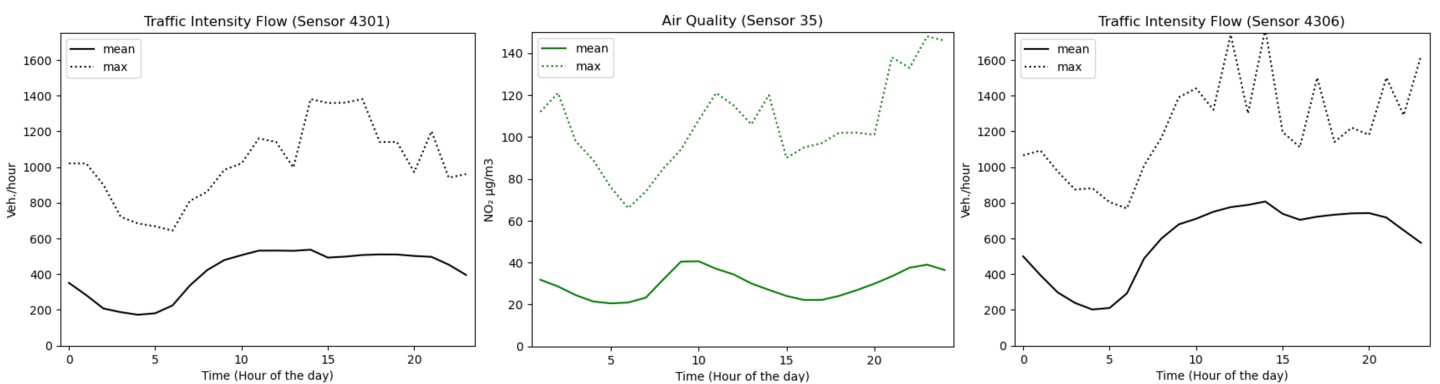

**Figure 8 Correlation between traffic intensity and air quality with respect to each hour of the day.**

values depending on the local calendar. As illustrated in Fig. 7, holidays affect the number of vehicles passing by both sensors. On the one hand, bank holidays that fall on Mondays have higher values than those for working Mondays at sensors 4301 and 4306. On the other, working Fridays are significantly higher than holiday Fridays. Both sensors provide similar behavior for the other weekdays. Sensors 4301 and 4306 have higher values for working Tuesdays, Thursdays, Fridays and Saturdays, but the absolute values on weekdays at sensor 4306 are higher. This shows the importance of this feature as regards traffic intensity distribution.

The last analysis is related to the importance of traffic intensity in the case of air quality levels. As shown in Fig. 8, traffic intensity affects air quality conditions. We focused on $NO_2$, since this is the parameter that is taken into account when implementing traffic restrictions in the city of Madrid (station 35). $NO_2$ behaves similarly to traffic intensity. The peak of vehicles/hour and gas levels detected by sensors 4301 and 4306 takes place between 10 am and 3 pm. Moreover, a valley is observed for both of them between 5 pm

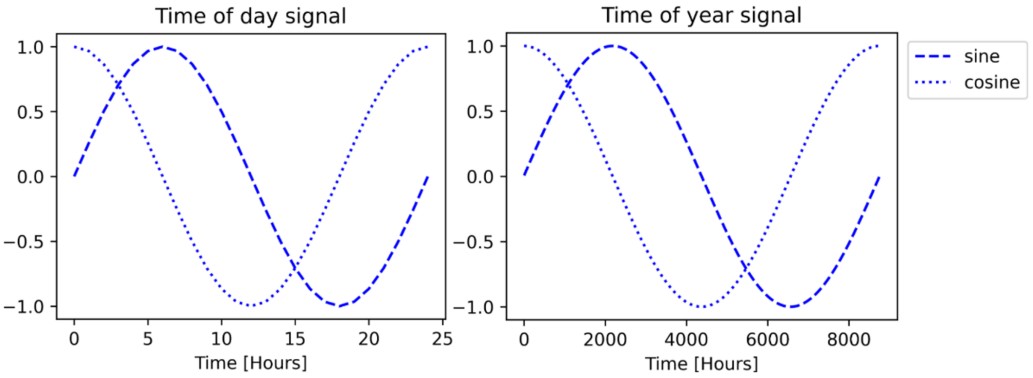

**Figure 9 Sinusoidal function for time day/year signal.**

and 10 pm. From 3 pm onwards, another significant increase appears that affects both the traffic intensity and the air quality. The max values are very similar for traffic intensity and air quality, with the exception of sensor 4301 for which there is a significant decrease in the last part of the day.

In our statistical-based feature selection analysis was used as the basis on which to consider the following features in our model: *Date timestamp, Hour, Weekday, Holiday, NO₂, Temperature, Humidity, Traffic Occupancy*, and *Traffic Intensity*. The only temporal features that do not have a regular periodicity are the months (28–31 days) and years (365–366 days) that can vary over time their values in contrast to hours (24). The management of periodic features demands to control the management of the inputs of the algorithm. Therefore, for objects that exhibit periodic behavior, a sinusoidal function can be used as a modeller since these functions are periodic. These functions solve the curse of dimensionality and longer computational time for machine learning algorithms (*Peng et al., 2021*).

As illustrated in Fig. 9, we applied this function for time day/year signal due to their periodicity. We calculated the sine as $\{sin((x) * (2 * \Pi / (day \vee year)))\}$ and the cosine as $\{cos((x) * (2 * \Pi / (day \vee year)))\}$, both for every 'date_timestamp' in our merged dataset (Data Layer) since 1st January of 2022. Therefore, these new features have been included in our model, instead of the original 'date_timestamp' field (Table 2).

As can be observed in Fig. 10, the relationship between the features varies. A heatmap shows the correlation between features, where the clearer the value between two features, the more correlated they are with values closer to 1. Using pollution levels as the basis for our study, intensity and occupancy are the most highly correlated. This validates the previous analysis where changes in pollution levels were reflected depending on traffic. Furthermore, hours have a significant effect for both days and years. By contrast, temperature has a low correlation with pollution levels.

### Predictive analysis

**Multi-step algorithm benchmarking (1):** Before running an algorithm (Fig. 11), it is common to split a dataset into training and testing sets before fitting a statistical or machine learning model (*Joseph, 2022*). We applied a (80%, 10%, 10%) split for the

**Table 2  Features considered in order to train the model.**

| Feature | Label | Value/unit |
| --- | --- | --- |
| Hour | hour_of_day | 0–23 |
| Time of day | Day sin & Day cos | [−1,1][a] |
| Time of year | Year sin & Year cos | [−1,1][b] |
| Weekday | day_of_week | 1–7 |
| Holiday | is_holiday | 0–1 |
| $NO_2$ | no2_level | mg/$m^3$ |
| Temperature | Temperature | °C |
| Humidity | Humidity | % |
| Traffic occupancy | Occupancy | 0–100 |
| Traffic intensity | Intensity | Veh/h |

**Notes:**
[a] Day (seconds) = 24 (h/day) * 60 (min/h) * 60 (sec/min).
[b] Year (seconds) = 365.25 (days/year) * day (sec).

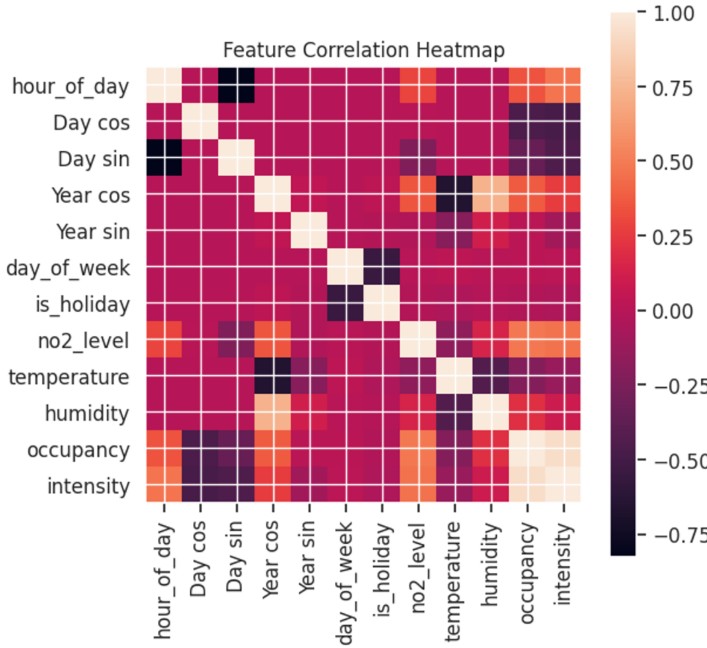

**Figure 10  Correlation between features (labels).**

training, validation, and test sets. Therefore, we split in such a way as to respect the temporal ordering and the model is never trained on data from the future and only tested on data from the future. Secondly, it is necessary to normalise the data in order to guarantee their quality and the accuracy of the machine/deep learning model (*Awan, Minerva & Crespi, 2020b*). We used one of the most common normalisation technique: Subtract the mean and divide by the standard deviation of each feature. In this technique, the mean and standard deviation should only be computed using the training data so that the models have no access to the values in the validation and test sets (*Aksu, Güzeller & Eser, 2019*). These features might have had disparate impacts on the quality of our model

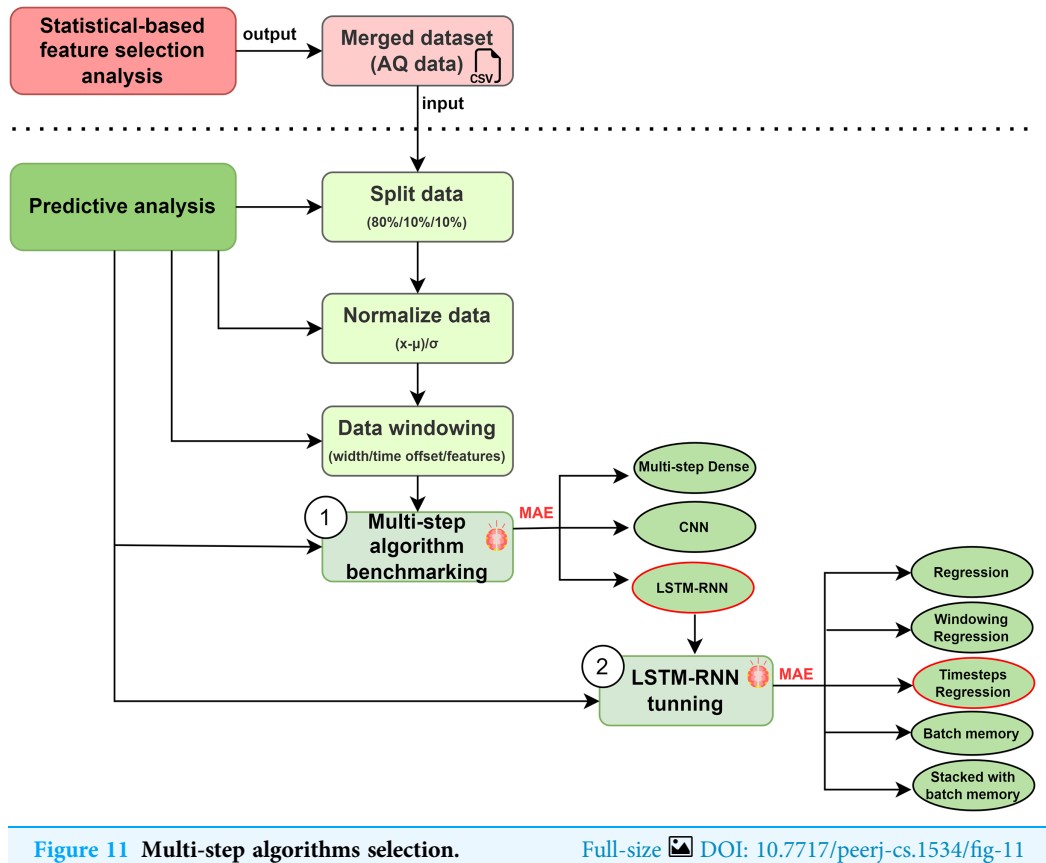

**Figure 11  Multi-step algorithms selection.**  

and we, therefore, studied the importance of each feature considered in order to estimate these possible impacts. Thirdly, the implementation of sliding window time series (SWTS) analysis enables to use the resulting predictor, but the driving data should be presented in the state-space format (*Mozaffari, Mozaffari & Azad, 2014*).

We used multi-step algorithms in order to work with dynamic and continuous data (*Belgasmia, 2021*). This approach is applied to discover the best algorithm for our prediction case, we compared the multi-step algorithms most frequently used in the state of the art: Multi-step Dense, LSTM-RNN and CNNs. Some approaches apply Multi-step Dense to classification problems by training them in a supervised manner; LSTM-RNN in time series problems; and CNNs in order to learn multiple layers of feature hierarchies automatically (*Ordóñez & Roggen, 2016*). The selection was determined by the lowest Mean Absolute Error (MAE) (*Šter, 2013*). We initialized the algorithms with two different strategies based on split sizes (Table 3).

**LSTM-RNN tuning (2):** As illustrated in Fig. 11, the best option as regards dealing with temporal patterns is the Long Short-Term Memory Recurrent Neural Network (LSTM-RNN) (*Guo et al., 2016*). In accordance with our model, the input will be expressed as $\{x = (x_1, x_2, ..., x_T)\}$ and the output as $\{y = (y_1, y_2, ..., y_T)\}$ with $\{T\}$ as time in our traffic intensity predictions, following the algebraic expressions shown in *Ma et al. (2015)*. The LSTM-RNN framework works with memory blocks, unlike traditional neural networks, which work with neurons. A memory block has an input gate, an output gate, and a forget

**Table 3  Multi-step algorithm benchmarking configuration.**

| Metric | Description | Strategy-1 | Strategy-2 |
|---|---|---|---|
| lag | Number of lags (hours back) to use for models | 168 | 168 |
| n_ahead | Steps ahead to forecast | 24 | 24 |
| split_size | Split share in testing | 80/10/10 | 70/20/10 |
| epochs | Epochs for training | 15 | 15 |
| batch_size | Batch size | 512 | 512 |
| lr | Learning rate | 0.001 | 0.001 |
| n_layer | Number of neurons in layer | 3 | 3 |

**Table 4  LSTM-RNN tuning configuration.**

| Metric | Strategy-1 | Strategy-2 | Strategy-3 | Strategy-4 | Strategy-5 | Strategy-6 | Strategy-7 | Strategy-8 |
|---|---|---|---|---|---|---|---|---|
| lag | 168 | 168 | 168 | 168 | 168 | 168 | 168 | 168 |
| n_ahead | 24 | 24 | 24 | 24 | 24 | 24 | 24 | 24 |
| split_size | 70/20/10 | 70/20/10 | 70/20/10 | 70/20/10 | 80/10/10 | 80/20/10 | 80/10/10 | 80/10/10 |
| epochs | 15 | 30 | 15 | 30 | 15 | 30 | 15 | 30 |
| batch_size | 512 | 512 | 512 | 512 | 512 | 512 | 512 | 512 |
| lr | 0.001 | 0.001 | 0.001 | 0.001 | 0.001 | 0.001 | 0.001 | 0.001 |
| n_layer | 4 | 4 | 8 | 8 | 4 | 4 | 8 | 8 |

gate. The output of the input gate is represented as $\{i_t\}$, the output gate as $\{o_t\}$, and the forget gate as $\{f_t\}$. Moreover, the cell and memory activation vectors are symbolized as $\{c_t\}$ and $\{m_t\}$, with $\{b\}$ and $\{W\}$ representing the bias and the weight. The centered logistic sigmoid functions are $\{h\}$ and $\{g\}$. Nevertheless, the lack of connection among neurons within the same layer has some limitations and increases the failure ratio in terms of spatio-temporal reasoning (*Zhao et al., 2017*).

To carry out this analysis, we defined five types of algorithm optimizations: Regression, Windowing regression, Timesteps regression, Batch memory, and Stacked batch memory. Accordingly, it is necessary to do a deeper study on the optimization of the algorithm: LSTM-RNN tuning. In a more detailed explanation of each optimization, Regression is the base calculation methodology for predictions. Windowing regression is intended to emulate the problem so that multiple recent steps can be used to make the prediction for the next time step. Timesteps regression takes previous time steps in our time series as inputs to predict the output at the next time step. Batch memory builds states throughout the entire training sequence and even maintains them if necessary to make predictions and lastly, Stacked batch memory which is the addition to the configuration that is required is that an LSTM layer before each subsequent LSTM layer must return the sequence for the predictions. To each of them we applied a series of strategies that are detailed below (Table 4).

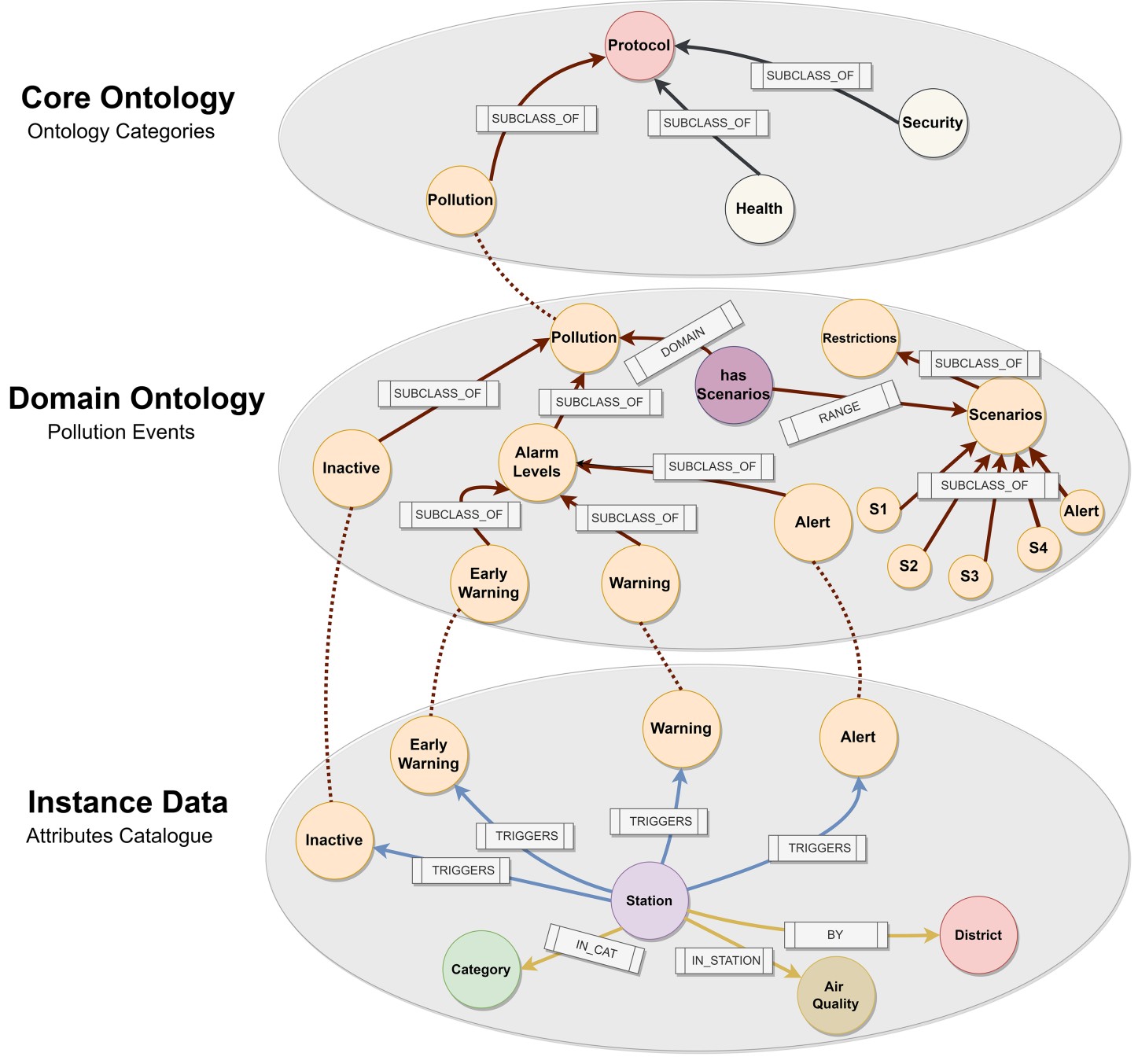

**Figure 12** Pollution protocol domain ontology.

### Event generation layer

The objective of this layer is to translate the Madrid City Pollution Protocol (*Madrid City Council, 2022*) into our own intelligence in order to generate events according to $NO_2$ levels. As shown in Fig. 12, we have designed an event-based ontology in order to express this logic semantically. We analysed other works that established ways to adapt the logic of

**Algorithm 1** Ontology inference sink Query.

```
MATCH  (high:Class { name: ?NO2_activation_levels})
CALL  n10s.inference.nodesInCategory(high, { inCatRel: "TRIGGERS" })
yield node AS priority
WITH priority MATCH (priority)-[:BY]->(b:Brand)
SET priority.itemLevel = ?NO2_activation_levels
return priority.itemName AS Station, b.brandName AS District,
priority.itemLevel AS AlarmLevel, priority.timestamp AS Datetime
```

smart cities within decision making. For this reason, we based the design of our ontology on the semantic expression works analysed in the 'Related Works'.

We first studied different protocols and activation plans. This process is helpful as regards defining the 'Core Ontology' hat contains the main class Protocol with sub-classes such as Pollution, Health or Security. All cities have different protocols for specific areas, and this level is where they relate to each other. We then defined the Protocol 'Domain Ontology', which shows the different $NO_2$ activation levels as *Early Warning, Warning, and Alert*. Early Warning activates when values exceed 180 mg/$m^3$, Warning 200 mg/$m^3$, and Alert 400 mg/$m^3$. In addition, pollution has different scenarios depending on air quality levels and number of times that any alarm has been activated. Finally, we connected our logic with the lowest level of the ontology. This level is denominated as 'Instance Data', where our ontology infers the predictive data. The relationship between the concepts is through the alarms that are generated based on pollution levels and the stations. The station has a category associated with it and belongs to a specific district within the city of Madrid. An external file (csv) contains all the fields required in order to activate any level (*datetime, url, itemId, itemName, brandName, itemStation, itemCategory and itemLevel*). We run a python script that imports this file in our graph database. Our Domain Ontology was created in OWL format (*World Wide Web Consortium (W3C), 2022*) and stored in an NEO4J graph database (*Neo4j, 2022*) to carry out this approach. This approach has already been used in *Gong et al. (2018)*.

## Ontology inference

In Algorithm 1, we use our ontology to infer pollution data in order to search for semantics. The result generates an event that triggers an alarm if any of the $NO_2$ level thresholds are exceeded (activation levels). This alarm allows us to anticipate increases in pollution and its effect on the city. At the time of running the inference, we must form a message with the date, station and pollution levels. What the ontology returns is an alarm based on whether the limits set by the protocol are exceeded, along with the datetime, station, and district. We run a python script with the inference sink query in order to generate an output. This output can be visualized in the Neo4J dashboard and at the same datetime, the result is sent to a Kafka topic.

**Table 5  Multi-step algorithm benchmarking experiments.**

| LSTM-RNN tunning | | | Configuration | | | Metrics | | | | | |
|---|---|---|---|---|---|---|---|---|---|---|---|
| | | | | | | RMSE | | MSE | | MAE | |
| Experiment | Algorithm | Strategy | Epochs | Layers | Splits | Train | Test | Train | Test | Train | Test |
| 1 | Multi-step D. | 1 | 15 | 3 | 70/20/10 | 31.16 | 30.45 | 9.71 | 9.27 | 28.22 | 27.94 |
| 2 | CNN | 1 | 15 | 3 | 70/20/10 | 29.32 | 28.67 | 8.59 | 8.21 | 26.59 | 25.94 |
| 3 | LSTM-RNN | 1 | 15 | 3 | 70/20/10 | 27.43 | 26.76 | 7.52 | 7.16 | 25.89 | 24.77 |
| 4 | Multi-step D. | 2 | 15 | 3 | 80/10/10 | 28.32 | 27.91 | 8.02 | 7.78 | 27.32 | 26.13 |
| 5 | CNN. | 2 | 15 | 3 | 80/10/10 | 26.94 | 26.74 | 7.25 | 7.15 | 24.58 | 23.13 |
| **6** | **LSTM-RNN** | **2** | **15** | **3** | **80/10/10** | **25.32** | **25.02** | **6.41** | **6.26** | **22.11** | **21.23** |

Note:
   Bold entries refer to the strategy with the best performance metrics and the highest values per metric.

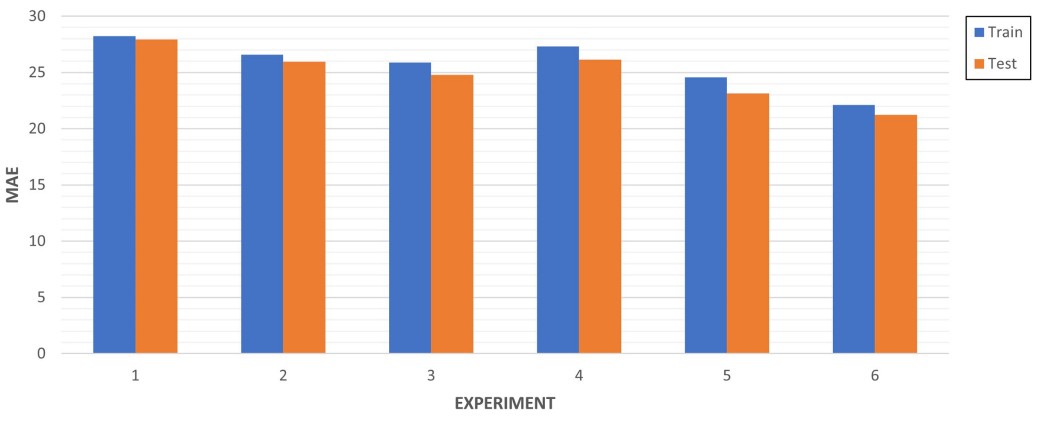

**Figure 13  Multi-step algorithms experiments.**     

## RESULTS

**Multi-step algorithm benchmarking** (Predictive analysis): In Table 5, we collected the results of algorithms benchmarking. LSTM-RNN ('LSTM') has the best performance when compared to CNN ('Conv') and Multi-Step Dense in both strategies. These two strategies provided different results depending on the dataset split strategy. This indicates that there is a better performance of any metric (MAE, MSE, and RMSE) increasing 10% the train data. From this analysis it can be deduced that the LSTM-RNN algorithm together with a strategy of 80%/10%/10% allows to obtain the best performance. Therefore, this is the base configuration of our model before its optimization.

As illustrated in Fig. 13, the more we increase the number of training data instances, the better results are obtained. Aligned with what was explained in Table 6, the experiment 6 is the one with the best performance.

In Fig. 14, we compared the top three experiments through its metrics (MAE, MSE and RMSE).

Experiments 5 and 3 have very similar values close to 27 in the RMSE, while experiment 6 improves both performances in this metric. This indicates the absolute fit of the model to

**Table 6 LSTM-RNN tuning experiments.**

| LSTM-RNN tuning | | | Configuration | | | Metrics | | | | | | |
|---|---|---|---|---|---|---|---|---|---|---|---|
| | | | | | | RMSE | | MSE | | MAE | |
| Experiment | Algorithm | Strategy | Epochs | Layers | Splits | Train | Test | Train | Test | Train | Test |
| 1 | Regression | 1 | 15 | 4 | 70/20/10 | 11.25 | 11.07 | 1.26 | 1.22 | 6.96 | 7.30 |
| 2 | Windowing R. | 1 | 15 | 4 | 70/20/10 | 11.15 | 10.96 | 1.24 | 1.20 | 6.99 | 7.18 |
| 3 | Timesteps R. | 1 | 15 | 4 | 70/20/10 | 11.51 | 11.42 | 1.32 | 1.30 | 7.08 | 7.61 |
| 4 | Batch m. | 1 | 15 | 4 | 70/20/10 | 12.72 | 11.68 | 1.61 | 1.36 | 9.67 | 8.81 |
| 5 | Stacked batch m. | 1 | 15 | 4 | 70/20/10 | 13.83 | 12.27 | 1.91 | 1.50 | 10.65 | 9.37 |
| 6 | Regression | 2 | 30 | 4 | 70/20/10 | 11.22 | 10.95 | 1.25 | 1.19 | 7.39 | 7.41 |
| 7 | Windowing R. | 2 | 30 | 4 | 70/20/10 | 11.24 | 11.11 | 1.26 | 1.23 | 6.88 | 7.27 |
| 8 | Timesteps R. | 2 | 30 | 4 | 70/20/10 | 11.17 | 11.00 | 1.24 | 1.21 | 6.83 | 7.21 |
| 9 | Batch m. | 2 | 30 | 4 | 70/20/10 | 12.72 | 11.67 | 1.61 | 1.36 | 9.59 | 8.76 |
| 10 | Stacked batch m. | 2 | 30 | 4 | 70/20/10 | 12.06 | 10.98 | 1.45 | 1.20 | 8.96 | 8.00 |
| 11 | Regression | 3 | 15 | 8 | 70/20/10 | 11.17 | 11.00 | 1.24 | 1.21 | 6.88 | 7.17 |
| 12 | Windowing R. | 3 | 15 | 8 | 70/20/10 | 11.23 | 11.08 | 1.26 | 1.22 | 6.85 | 7.21 |
| 13 | Timesteps R. | 3 | 15 | 8 | 70/20/10 | 11.57 | 11.39 | 1.33 | 1.29 | 7.64 | 7.71 |
| 14 | Batch m. | 3 | 15 | 8 | 70/20/10 | 12.92 | 11.91 | 1.67 | 1.41 | 9.63 | 8.90 |
| 15 | Stacked batch m. | 3 | 15 | 8 | 70/20/10 | 12.54 | 11.45 | 1.57 | 1.31 | 9.14 | 8.37 |
| 16 | Regression | 4 | 30 | 8 | 70/20/10 | **11.14** | 11.00 | **1.24** | 1.21 | **6.80** | 7.11 |
| 17 | Windowing R. | 4 | 30 | 8 | 70/20/10 | 11.24 | 11.08 | 1.26 | 1.22 | 7.30 | 7.42 |
| 18 | Timesteps R. | 4 | 30 | 8 | 70/20/10 | 11.08 | 10.87 | 1.22 | 1.18 | 6.82 | 7.07 |
| 19 | Batch m. | 4 | 30 | 8 | 70/20/10 | 12.65 | 11.54 | 1.60 | 1.33 | 9.67 | 8.67 |
| 20 | Stacked batch m. | 4 | 30 | 8 | 70/20/10 | 12.36 | 11.08 | 1.52 | 1.22 | 9.01 | 7.99 |
| 21 | Regression | 5 | 15 | 4 | 80/10/10 | 11.40 | 9.77 | 1.30 | 0.95 | 7.02 | 6.88 |
| 22 | Windowing R. | 5 | 15 | 4 | 80/10/10 | 11.75 | 10.25 | 1.38 | 1.05 | 7.23 | 7.30 |
| 23 | Timesteps R. | 5 | 15 | 4 | 80/10/10 | 11.73 | 10.13 | 1.37 | 1.02 | 7.82 | 7.32 |
| 24 | Batch m. | 5 | 15 | 4 | 80/10/10 | 13.21 | 10.59 | 1.74 | 1.12 | 10.14 | 8.39 |
| 25 | Stacked batch m. | 5 | 15 | 4 | 80/10/10 | 12.31 | 10.17 | 1.51 | 1.03 | 9.30 | 7.84 |
| 26 | Regression | 6 | 30 | 4 | 80/10/10 | 11.65 | 9.94 | 1.35 | 0.98 | 7.64 | 7.07 |
| 27 | Windowing R. | 6 | 30 | 4 | 80/10/10 | 11.43 | 9.68 | 1.30 | 0.93 | 7.47 | 6.95 |
| **28** | **Timesteps R.** | **6** | **30** | **4** | **80/10/10** | 11.30 | **9.58** | 1.27 | **0.91** | 7.11 | **6.76** |
| 29 | Batch m. | 6 | 30 | 4 | 80/10/10 | 13.17 | 10.69 | 1.73 | 1.14 | 10.33 | 8.48 |
| 30 | Stacked batch m. | 6 | 30 | 4 | 80/10/10 | 12.33 | 10.11 | 1.52 | 1.02 | 9.39 | 7.84 |
| 31 | Regression | 7 | 15 | 8 | 80/10/10 | 11.47 | 9.71 | 1.31 | 0.94 | 7.54 | 7.00 |
| 32 | Windowing R. | 7 | 15 | 8 | 80/10/10 | 11.47 | 9.91 | 1.31 | 0.98 | 7.04 | 7.02 |
| 33 | Timesteps R. | 7 | 15 | 8 | 80/10/10 | 11.64 | 10.24 | 1.35 | 1.04 | 7.24 | 7.38 |
| 34 | Batch m. | 7 | 15 | 8 | 80/10/10 | 12.37 | 10.17 | 1.53 | 1.03 | 9.13 | 7.83 |
| 35 | Stacked batch m. | 7 | 15 | 8 | 80/10/10 | 12.65 | 10.45 | 1.60 | 1.09 | 9.76 | 8.04 |
| 36 | Regression | 8 | 30 | 8 | 80/10/10 | 11.43 | 9.81 | 1.30 | 0.96 | 7.32 | 7.10 |
| 37 | Windowing R. | 8 | 30 | 8 | 80/10/10 | 11.54 | 9.88 | 1.33 | 0.97 | 7.38 | 7.00 |
| 38 | Timesteps R. | 8 | 30 | 8 | 80/10/10 | 11.51 | 10.04 | 1.32 | 1.01 | 7.11 | 7.19 |
| 39 | Batch m. | 8 | 30 | 8 | 80/10/10 | 11.96 | 9.97 | 1.43 | 0.99 | 8.65 | 7.51 |
| 40 | Stacked batch m. | 8 | 30 | 8 | 80/10/10 | 12.31 | 10.24 | 1.51 | 1.04 | 9.38 | 7.96 |

**Note:**
Bold entries refer to the strategy with the best performance metrics and the highest values per metric.

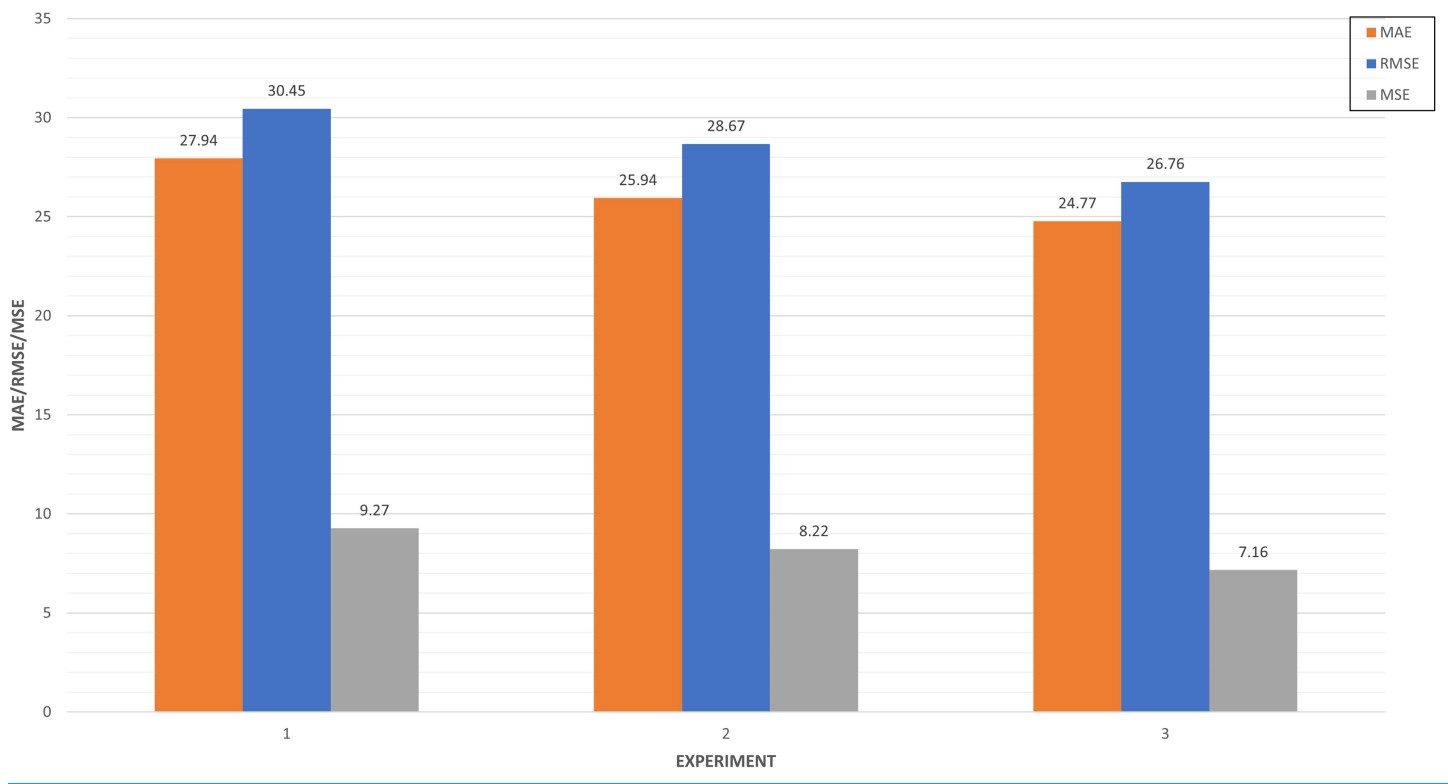

**Figure 14 Top three algorithm experiments.**               

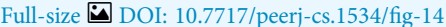

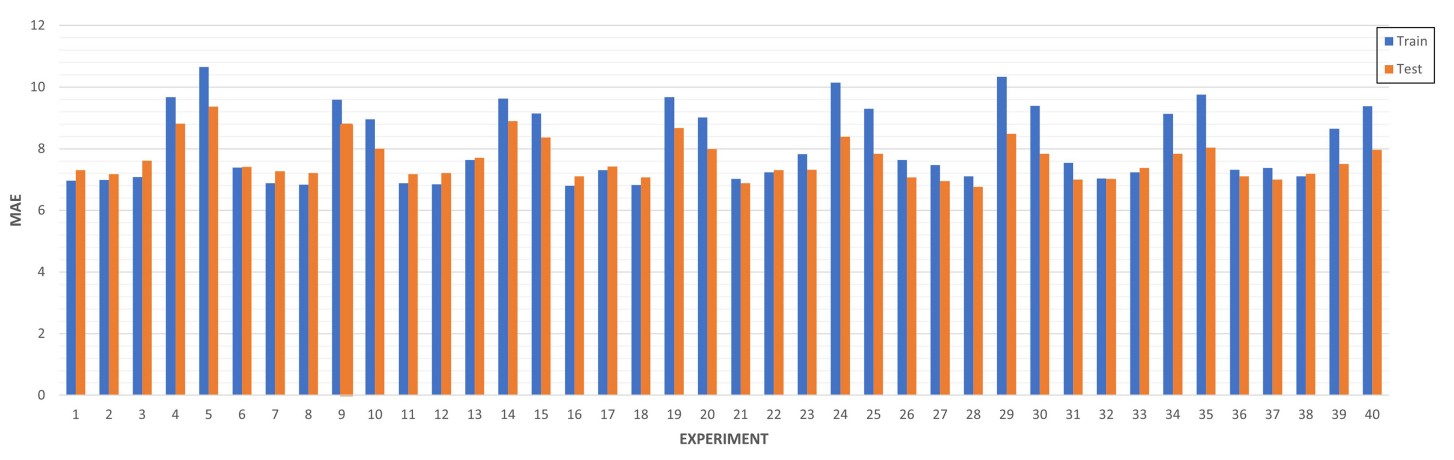

**Figure 15 Algorithm tuning comparison by experiment.**     

the data, how close the observed data points are to the predicted values of the model. In the case of MAE, the experiment is the only one that approaches 20, which represents the mean absolute error between the predictions of the training and the test.

LSTM-RNN Forecast (Predictive analysis): As evidenced in Fig. 15, most of the experiments are between the range of 6 and 8 of MAE. It can be ensured that the

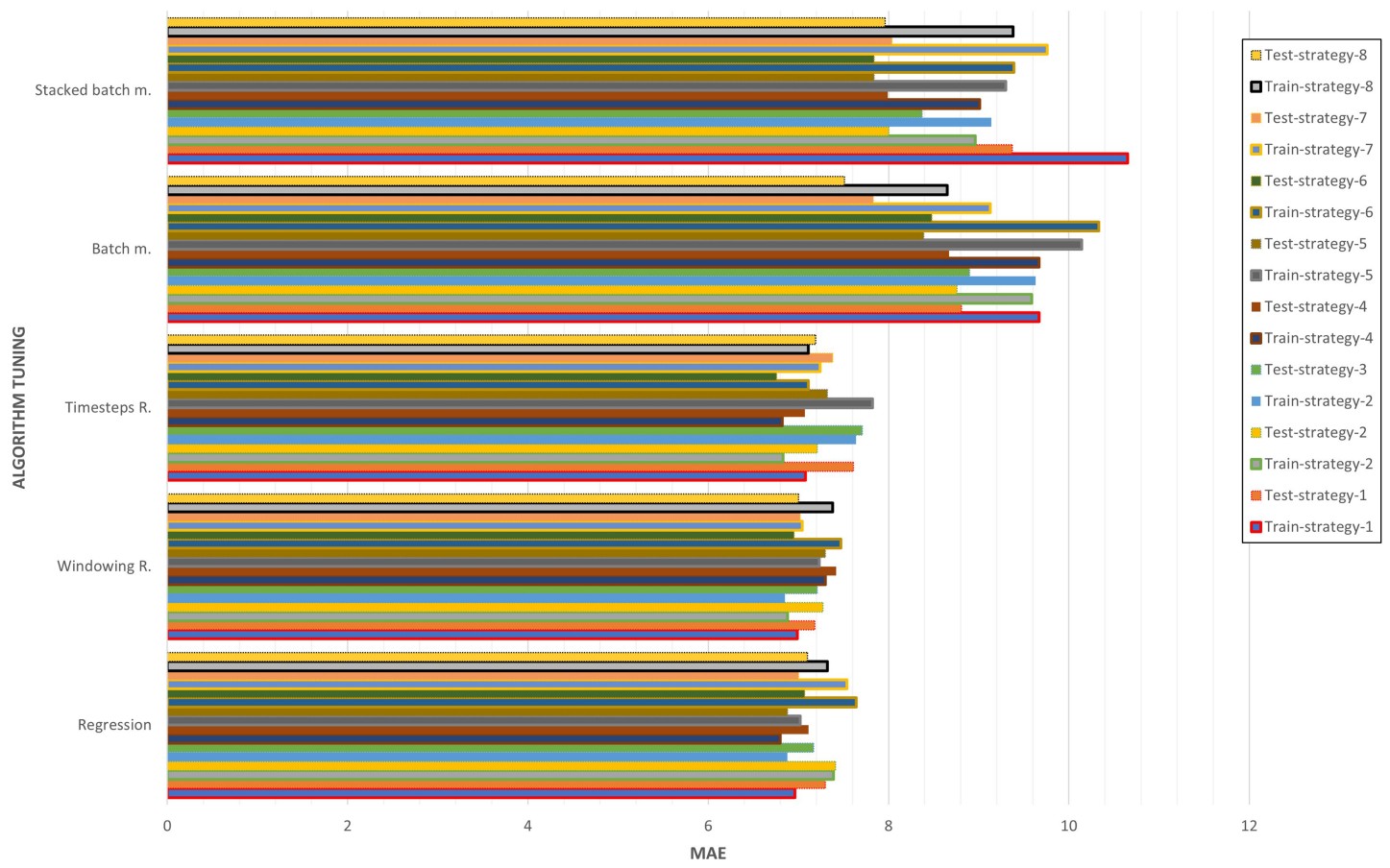

**Figure 16** Algorithm tuning comparison by strategy.

experiments that have a higher training MAE have worse performance at the test level. In addition, the models are better in those where the MAE of train improves that of test. In the other cases, there is surely underfitting and the model does not generalize well.

We collected experiments results in Table 6. A total of forty experiments have been carried out combining the eight strategies defined based on splits, epochs and layers.

As shown in Fig. 16, the best algorithms have been selected based on the strategy. It can be seen how the optimizations through regression, windowing regression and timesteps regression have values less than 8 of MAE for both the training and the test. On the other hand, the optimizations through batch memory and stacked batch memory exceed in at least one of them at the threshold of 8, and even 10 of MAE in some cases. In general, it can be concluded that strategies 5 and 6 are the ones that have obtained the best results for all the experiments. This is because the data has been increased by 10% and that significantly influences performance.

In Fig. 17, we compared the top five experiments through its metrics (MAE, MSE and RMSE). Experiments 27 and 31 have very similar values close to 10 in the RMSE, while experiment 28 improves both performances in this metric. This means the squared

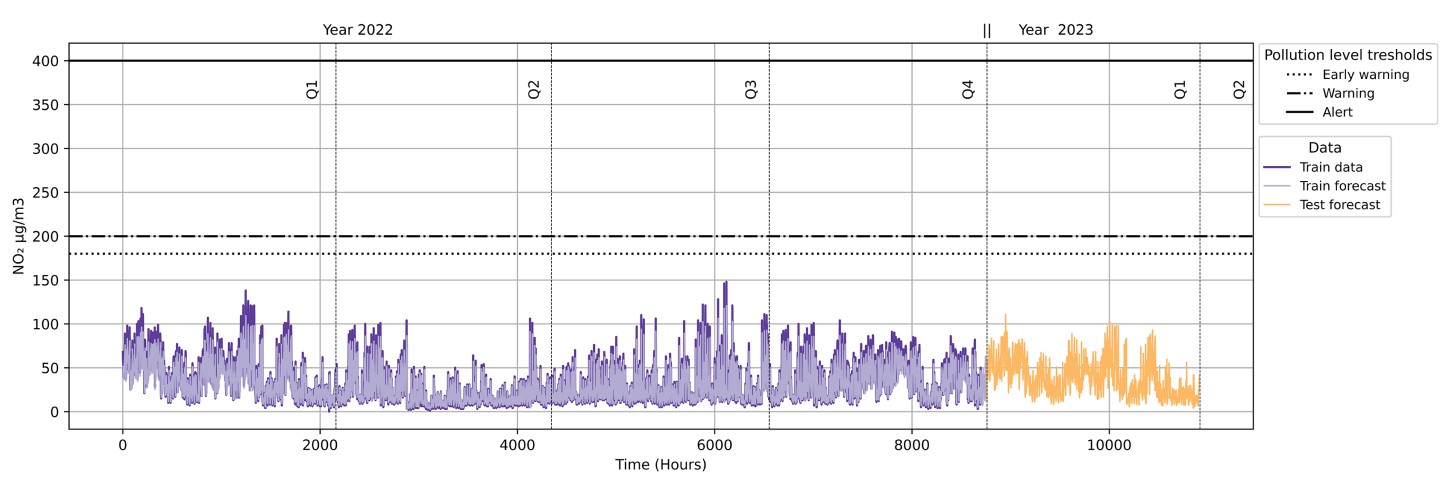

**Figure 17 Top five algorithm tuning experiments.**

**Figure 18 NO$_2$ level with pollution protocol levels in Madrid Central district.**

difference between the predictions and the target and then averages those values. In the case of MAE, five experiments are around seven, which illustrates the mean absolute error between the predictions of the training and the test.

**Ontology inference** (Event Generation Layer)

As shown in Fig. 18, the quarterly forecast does not trigger any alarms originating from pollution protocol levels. This implies that no traffic restrictions will be activated because the $NO_2$ levels are not reaching the minimum of any of the pollution protocol levels (Early Warning, Warning and Alert).

As shown in Fig. 19, our ontology was used to infer $NO_2$ level data in order to discover the location and the active pollution level. This figure illustrates how our instance data is linked to our domain ontology.

As explained in the definition of the Pollution Protocol Domain Ontology (see Event Generation Layer), the 'Domain Ontology' is represented with its alarm levels and established scenarios (orange color). These alarm levels have a defined range of $NO_2$ pollution in accordance with those established in the official protocol. On the other hand, in the 'Instance Data' we have the stations (light purple color) and their different attributes: category (green), air quality (brown), and district (red). The 'Instance Data' will be connected through the 'triggers' relationship in the event that one of its values at a given time is within the range established within an alarm level. This happens because each station provides a range of pollution levels according to the predictions. For instance, in this figure the relationship 'triggers' connects 'Station 35' to the alarm level 'Inactive' based on some datetimes in Q1 2023. In this case, no level remained active for these datetimes.

## DISCUSSION

Designing a dynamic application of traffic restrictions is based on connecting Pollution Level Alarms with different scenarios. Madrid City Council provides five scenarios depending on the alarm level and duration (*Madrid City Council, 2022*). Certain restrictions are applied in these scenarios, such as speed limits and access to districts:

- *'Scenario 1'* restricts the speed limit to 70 km/h.
- *'Scenario 2'* restricts the speed limit to 70 km/h, and access is authorised for 'Zero emission' ehicles (excluding taxis).
- *'Scenario 3'* restricts the speed limit to 70 km/h and access is authorised for 'Zero emission' ehicles (highly recommended for taxis).
- *'Scenario 4'* restricts the speed limit to 70 km/h and access is authorised for 'Zero emission' ehicles (recommended for taxis).
- *'Scenario Alert'* restricts the speed limit to 70 km/h and access is authorised for 'Zero emission' (including taxis).

As shown in Table 7, the alarm level and duration establish the requirements needed in order to activate any of the pollution protocol scenarios.

As illustrated in Fig. 18, neither the quarterly forecast data (Year 2023) nor the data in 2022 exceeded pollution alarm thresholds. In order to validate our approach, we carried

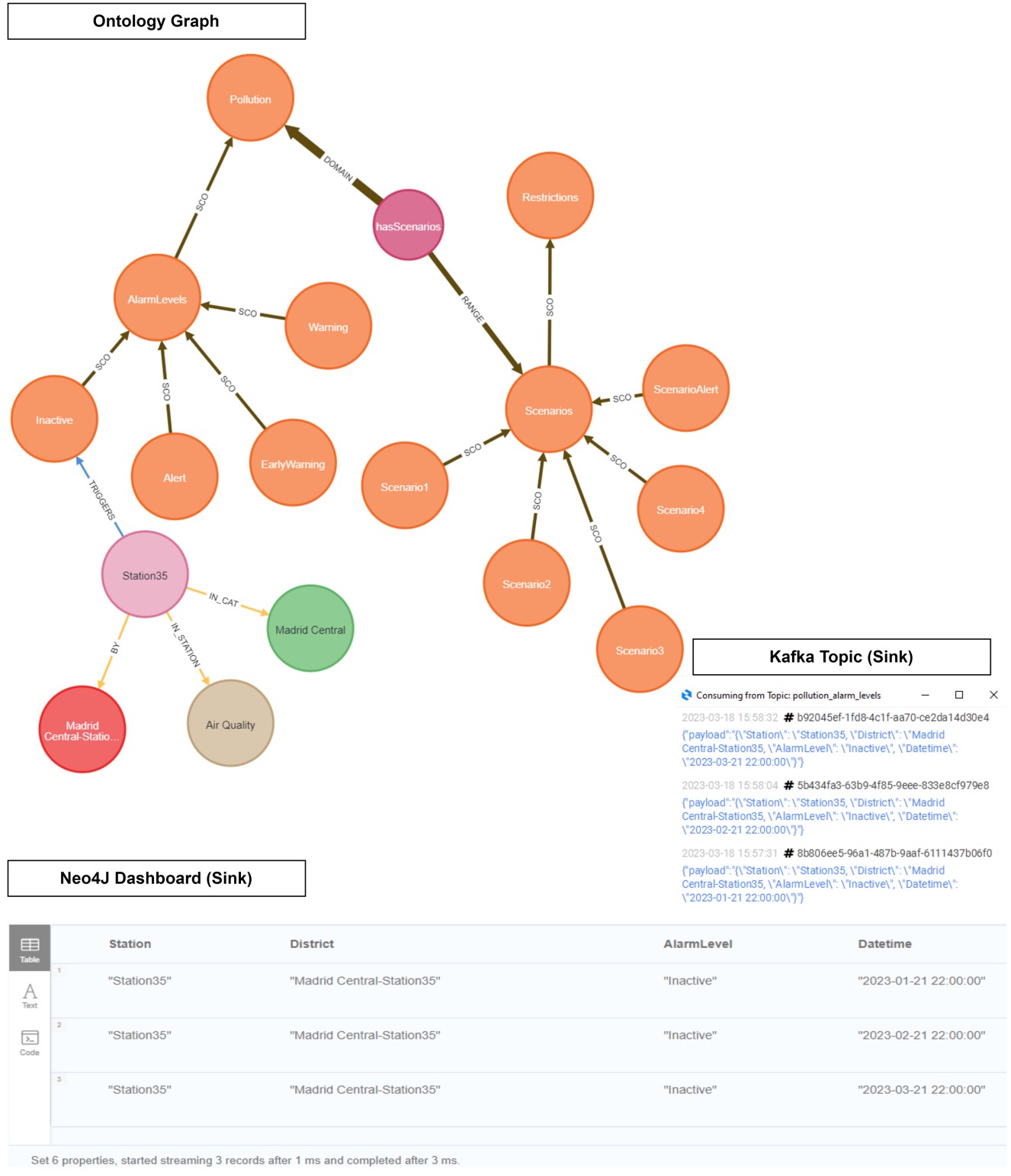

**Figure 19** Ontology inference $NO_2$ level in the Madrid Central district.

**Table 7 Pollution protocol scenarios.**

| Scenario | Level (duration) | Restrictions |
|---|---|---|
| 1 | Early warning (1 day) | Speed |
| 2 | Early warning (2 days)/ warning (1 day) | Restricted access and speed |
| 3 | Early warning (3 days)/ warning (2 days) | Restricted access and speed |
| 4 | Warning (4 days) | Moderate restricted access and speed |
| Alert | Alert (1 day) | Full restricted access and speed |

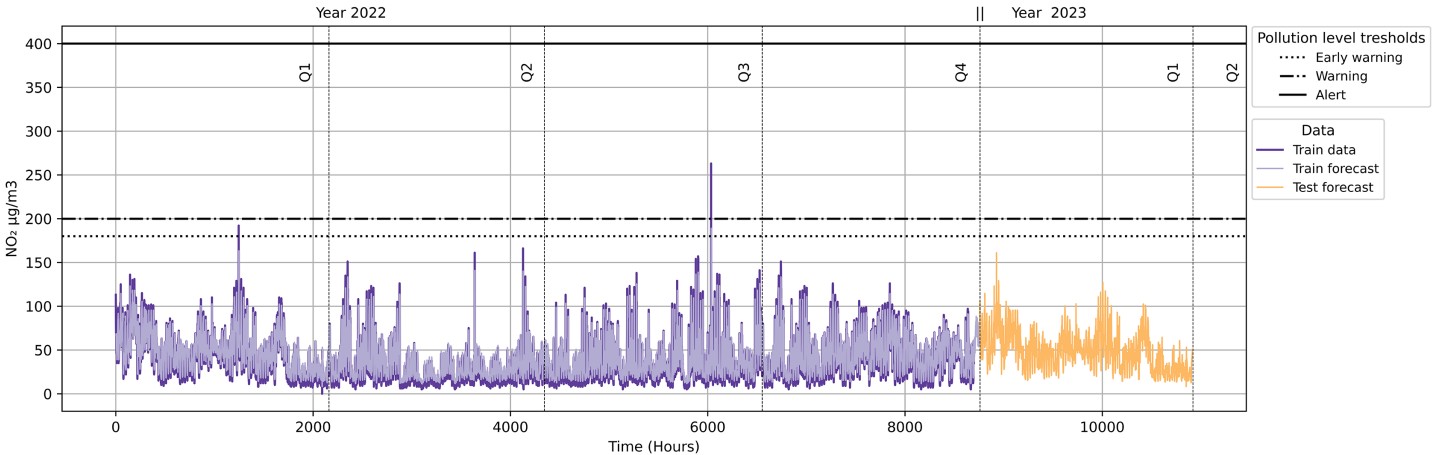

**Figure 20 NO$_2$ level with pollution protocol levels in the Salamanca district.**

out the same methodology in case the contamination protocols were expanded to other districts. For this, we used the Salamanca district as the basis for our validation. We compared this district with Madrid Central to analyse the possible differences in results at the same datetime.

As seen in Fig. 20, the datetime '2022-02-21 22:00:00' exceeded 'Early Warning' level, and the datetime '2022-09-09 11:00:00' a 'Warning' level. Other datetimes remained at the 'Inactive' level. The dates '2022-02-21' and '2022-09-09' would activate 'Scenario 2' of the Pollution Protocol. We then inferred this data to verify this scenario. We applied our validation with the datetime '2022-02-21 22:00:00' because it refers the first quarter.

As evidenced in Fig. 21, 'Station 35' (Madrid Central district) triggers an 'Inactive' level, while 'Station 8' (Salamanca district) triggers an 'Early Warning' level. We see how the output of the inference is displayed both in the Neo4J dashboard and within a Kafka topic.

Scenario 2 applies some restrictions related to speed limits and access to districts. The speed cannot exceed 70 km/h at district access points during the activation of this protocol

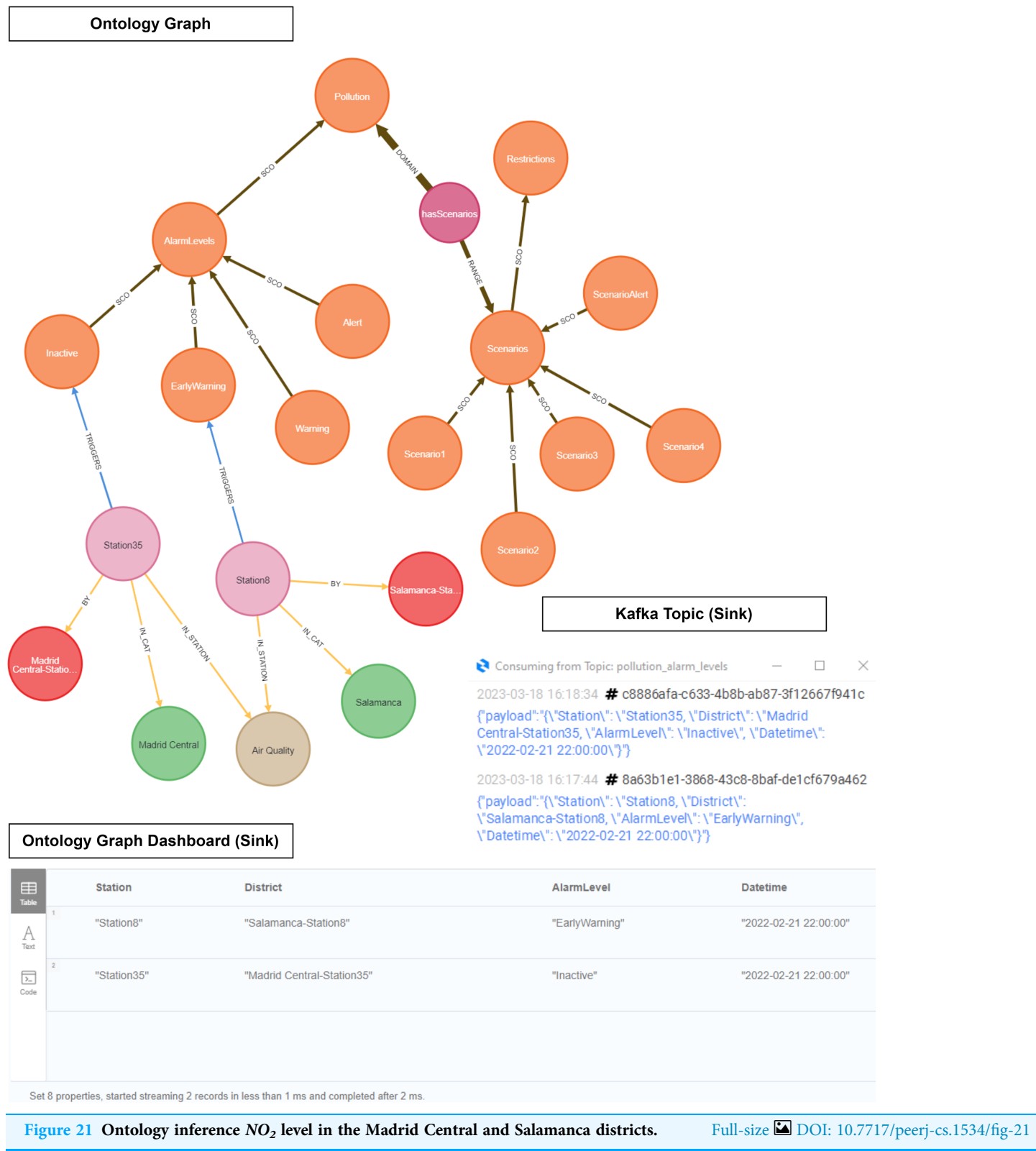

**Figure 21** Ontology inference $NO_2$ level in the Madrid Central and Salamanca districts.

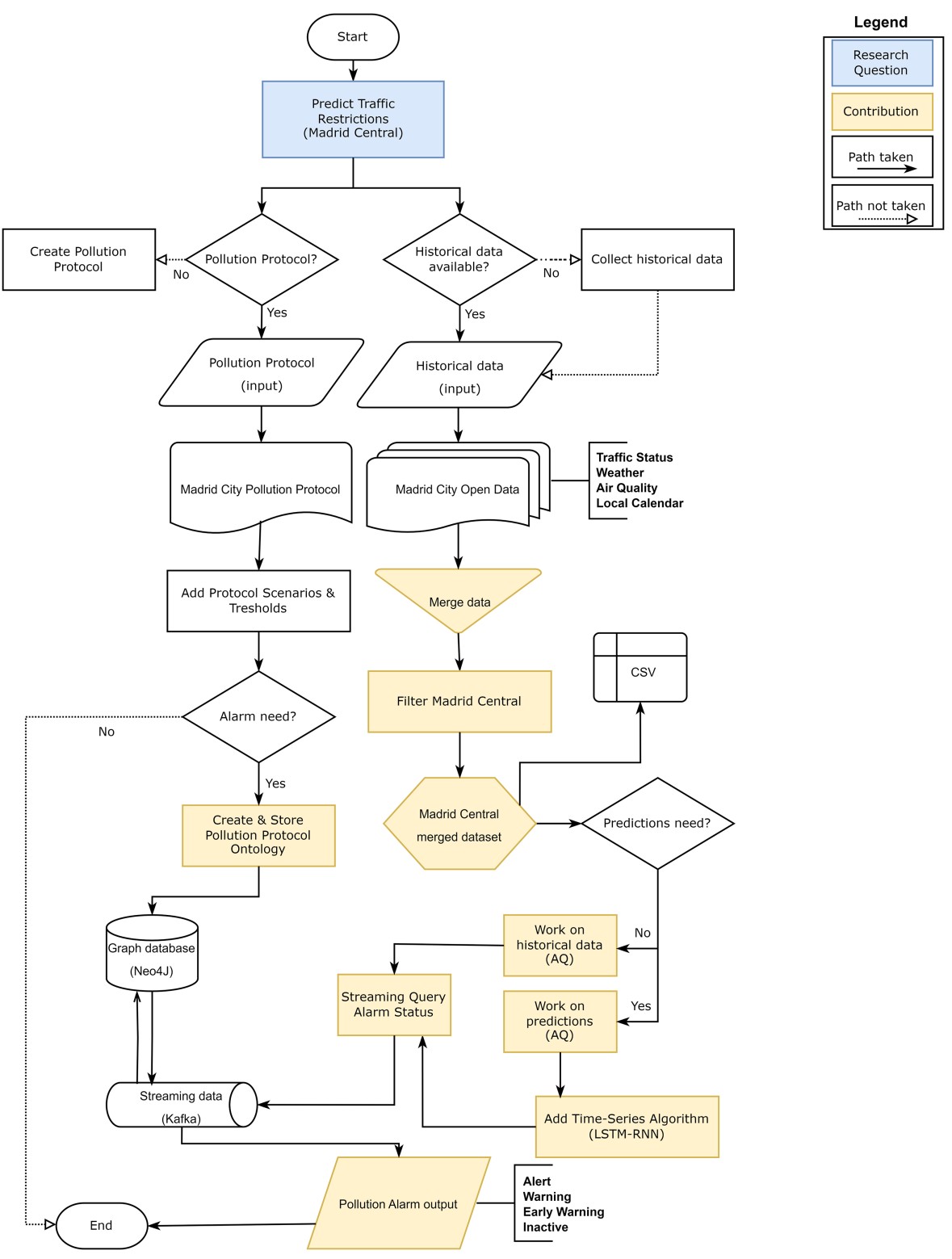

**Figure 22** Flowchart for the research validity check.

scenario. Moreover, driving without an environmental 'Zero emissions' label is prohibited. These restrictions are applied the day after the activation of the scenario.

It is consequently possible to confirm that our contamination protocol ontology works correctly as regards triggering a pollution level alarm. On the one hand, it has been able to transfer the pollution predictions to the first quarter of 2023, and on the other hand, it has allowed us to see how it would work with several stations at the same datetime.

As a final part of the discussion, we wanted to explain the validity of our approach through a flowchart (Fig. 22). Flowcharts serve as a pictorial means of communicating from one person to another the temporal order of events or actions through American National Standard Institute (ANSI) (*Chapin, 2003*).

Based on these standards, our validation process begins with the definition of the research question: Predict traffic restrictions. For this, we established Central Madrid as the validation scenario. On the one hand, we investigated if there is any public that limits the restrictions in this district: the protocol of political contamination. This protocol determines a series of scenarios based on thresholds based on air quality pollution levels. At this point we considered the need to generate a system that could launch alarms when the contamination levels established in the protocol were exceeded. This led us to create an ontology based on the contamination protocol and store it in a graph database (NEO4J) as the basis of the approach. On the other hand, in order to validate this approach, we needed to have historical data in order to work: Madrid City Open Data. We focused on generating a first dataset to merge the levels of traffic, weather, air quality and local calendar as a basis for our study. We then filtered by stations that were within the Madrid Central district boundaries to create a dataset. On this dataset, we defined the objective of working on predictions, but using historical data as part of the validation test. In order to make the predictions, we analysed the time series algorithms that could give the best results to select the one with the best metrics. At this point, we needed to join the ontology stored in our graph database (NEO4J) with the output data of the predictions and we established a streaming data management tool (Kafka). This tool had a connector with the database to launch queries that returned the alarm levels of the stations based on those predictions: Inactive, Early Warning, Warning and Alert.

## CONCLUSION

This work shows the impact of traffic on pollution levels. We used real data in order to estimate and predict future pollution level values. Moreover, we integrated the Madrid City Council pollution protocol scenarios into our ontology in order to automatically trigger alarms based on pollution threshold levels. This system provides advantages by which to include local pollution policies based on traffic predictions. This has been accomplished by proposing a framework with an additional layer of events that takes into account the variability of the specific prediction case for air pollution forecasting with local policies. Our approach employs this framework in order to overcome certain limitations indicated in literature. Authors stated the need for dynamic systems with which to manage real time traffic status based on pollution protocols (*Rodriguez Rey et al., 2021*). The approach presented in this work, therefore, triggers events/alarms on the basis of Madrid city

council's pollution protocol. Moreover, we included local calendar holidays or traffic occupancy, which have not been considered in previous research (*Borge et al., 2018*). We showed the effect of holidays and weekdays on traffic and pollution levels. We also justified the use of deep learning techniques in forecasting systems (*Pappalardo et al., 1998*), in addition to validating our ontology and the entire framework. This framework, therefore, has the potential to be dynamic and to adapt to what is happening in real time scenarios. This might be useful as regards extending traffic restriction policies and testing the effects in different pollution scenarios in the city, thus enabling local governments to comply with EU air quality standards in order to improve the quality of citizens' lives.

## Limitations

This work is focused on the 'Madrid Central' district, which has more restrictions than any other place in the city of Madrid. Nevertheless, this district has fewer air quality stations than other districts. We have, therefore, prioritised restrictions rather than a larger amount of data.

## Future work

In future work, we plan to extend our experiments to more protocols in order to understand the importance of developing ontologies. Furthermore, we intend to collect data for different districts and compare them.

### Funding

This work has been developed within the AETHER-UCLM (PID2020-112540RB-C42) funded by MCIN/AEI/10.13039/501100011033, ALBA-UCLM (TED2021-130355B-C31, id.4809130355-130355-28-521), funded by "Ministerio de Ciencia e Innovación" and MESIAS (2022-GRIN-34202) funded by FEDER. The funders had no role in study design, data collection and analysis, decision to publish, or preparation of the manuscript.

### Grant Disclosures

The following grant information was disclosed by the authors:
AETHER-UCLM (PID2020-112540RB-C42).
MCIN/AEI/10.13039/501100011033.
ALBA-UCLM (TED2021-130355B-C31, id.4809130355-130355-28-521).
"Ministerio de Ciencia e Innovación" and MESIAS: (2022-GRIN-34202).
FEDER.

### Competing Interests

The authors declare that they have no competing interests.

## Author Contributions

- David Eneko Ruiz de Gauna conceived and designed the experiments, performed the experiments, performed the computation work, prepared figures and/or tables, authored or reviewed drafts of the article, and approved the final draft.
- Luís Enrique Sánchez conceived and designed the experiments, prepared figures and/or tables, and approved the final draft.
- Almudena Ruiz-Iniesta analyzed the data, performed the computation work, authored or reviewed drafts of the article, and approved the final draft.

## Data Availability

Data are available at Zenodo:

David Eneko Ruiz de Gauna, Luis Enrique Sánchez, Almudena Ruiz-Iniesta, & Mireia Yurrita. (2022). Design of a pollution ontology-based event generation framework for the dynamic application of traffic restrictions (0.1). Zenodo. https://doi.org/10.5281/zenodo. 7405672.

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
