# Peer review of "Design of a pollution ontology-based event generation framework for the dynamic application of traffic restrictions"

_PeerJ Computer Science, doi:10.7717/peerj-cs.1534_

## Round 0.1 · original submission · Major Revisions

We acknowledge the good work done by the authors in improving the paper but some still work needs to be done especially on the experimental workflow, to clarify any ambiguities (see reviews).

·

Basic reporting

Please note that the row numbers refer to the tracked differences file.

There are some typos and misspellings.
- Table 1: "Section Prediction Layer" (capital letters)
- Phrase in rows 253-254 is difficult to understand
- There are some discrepancies between the names of the attributes as reported in the text, Table 2, Figure 9 and Figure 10
- Row 187: estimated ---> selected (?)
- Row 299: election ---> Selection (?)
- Row 318: Bath ---> Batch (?)
- Left quotation marks are not used correctly throughout the paper (always right quotation marks are used)
- Row 394: allows obtaining ---> allows to obtain
- Row 395: best metrics ---> best performance (?)
- Figure 7: it is not very indicative, and it could be replaced with a graph based on boxplots instead (just a suggestion)
- Figures 13, 14, 15, 16, 17: they do not add much information with respect to Tables 5 and 6. Anyway, they do not employ the right kind of visualization. For instance, line plots should not be used, since the information on the x-axis is categorical (not continuous)

Experimental design

Please note that the row numbers refer to the tracked differences file.

- Row 91: according to the answers to the reviewers, the available data should now encompass also year 2022. Still, here the authors report "the last full year with all the data for every month is 2021"
- Table 1: how comes that, on the merged dataset, the number of columns after the filtering process raise from 9 to 10? Also, in the text, 9 attributes are reported in lines 195-203, while 10 attributes are mentioned in lines 254-255.
- Row 170: what is the exact meaning of occupancy?
- Row 226: "At first glance, the lower 225 the temperature, the higher the number of vehicles passing by sensors 4301 and 4306" ---> this phrase is still misleading
- Table 2: isn't the "Hour" attribute the same as "Time in day", but without the trigonometric transformation? How would the authors justify keeping it?
- Row 272: it is not clear how the training/validation/set split has been performed. Did it take into account the temporal dimension? Or was it full-random? Between the two approaches, I would suggest the first one, especially if data regarding year 2022 is now available.
- MAE, MSE, RMSE are not in %, but they refer to absolute values. MAPE is the % version of MAE.

Validity of the findings

I still cannot evaluate the validity of the findings with enough confidence, given the results reported by the authors and the overall description of the experimental workflow.

Additional comments

While the paper has improved, I believe it still requires work from the authors.
Please, in revising the work focus especially on the experimental workflow, to clarify any ambiguities.

Reviewer 2 ·

Basic reporting

The manuscript presents a framework and case study of a system designed to capture data from multiple sensors positioned within an urban geographic area (in this case, central Madrid) and utilise this to generate warnings to determine whether or not traffic restrictions should be imposed. In short, the paper surveys the literature regarding environmental decision making (primarily, but not exclusively though the use of machine learning), and then discusses and evaluates a multi-layer framework for generating travel restrictions based on inferred solution levels within Madrid.

There has been a significant improvement in this paper; primarily with respect to the discussion of the learning approaches taken and the results, but in other areas too, such as an improved related work section and an improved characterisation of dataset used. The paper is now in a better shape, but a few clarifications / corrections should be addressed - see the additional comments below.

Experimental design

no comment

Validity of the findings

no comment

Additional comments

In general - please consider fixing the use of single quotes. Throughout you use a standard quote both at the begging and end of quoted material - use a back quote at the beginning of each quoted phrase instead
Line 82-83 “... we present an ontology…whose purpose is to design a methodology…” -> perhaps expand to say “…whose purpose is to facilitate the design of …”?
Line 91 “…employing an ontology based on neural networks…” -> would be good to expand on this - do you mean that the instantiation of the ontology is based on data generated by the neural networks?
Line 143-144 “…This layer ingests this data and its output is a forecast. This forecast is then inferred by the Event Generation Layer by means of our ontology…” -> This doesn’t make sense; the forecast is either generated by the ML algorithm or inferred by the ontology. But surely it can’t be both. Please clarify.
Table 1. -> it is unclear where this is discussed in the text
Line 210: “…in order to explain the features correlation…” -> “…in order to explain the correlation of the different features…” ???
Fig 9 - this doesn’t appear to be referenced in the text. Is this what you refer to in Lines 255-258? Please clarify the reason for using the sinusoidal functions - for example, the cosine function falls to -1 during mid day - but what is this referring to?
Line 258 “…in our model. (Table 2).”-> I think you need to remove the first period after `model’.
Figure 10 - Great figure. Please consider clarifying the heat map values in the caption - for example, “A heat map showing the correlation between features, where the lighter the value between two features, then the more correlated they are”.
Table 3 - please explain what the different metrics mean (or at least make this clearer)
Line 353 “…the more we increase the number of training data, the better…” -> either change `number’ to `size’ (“…the more we increase the size of the training data, the better…” ) or add the word `instances’ (“…the more we increase the number of training data instances, the better…” )
Lise 357-358 “This means the squared difference between the predictions and the target and then averages those values.” - consider rewriting as I don’t believe this makes sense.
Line 369-370 “the optimisations though batch memory and stacked batch memory exceed in at least one of them the threshold…” -> do you mean “at the threshold”?
Line 384 “This figure illustrates how our instance data is linked to our domain ontology” - it doesn’t really, I’m afraid/. You need to explain the figure a little more.
Line 402 “…Table 7,the…” - space before “the”
Line 404 “As shown in Fig 19” - do you really mean 19 (which illustrates the ontology instantiation)? If so, explain how it is shown.
Line 407 “We compared this Madrid Central to analyse” -> sentence just stops.
Line 409 “Other date times remained ‘Inactive’ level.” -> do you mean “Other date times remained at the `Inactive’ level.”???
Line 414 “We appreciate how the output of the inference is displaced both in the Neo4J dashboard ad within a Kafka topic.” -> I don’t think `appreciate’ is the right word here
Line 420 “It is consequently is possible” -> remove the second `is’

---

## Round 0.2 · Minor Revisions

Please carefully address the remaining comments of Reviewer 1 concerning the experimental design.

·

Basic reporting

There are some typos and misspellings.

- In table 1, the asterisk does not refer to any of che columns.
- On page 6: "ut is is necessary to study" ---> thus (?)
- On page 8: "dimension disaster problem" ---> I believe it should be "curse of dimensionality"?

Experimental design

- I still have concerns regarding Figure 5: for sensor 4306, it simply doesn't show that the traffic intensity increases as the temperature decreases
- In figure 7, I would at least try to scatter the dots a little bit adding some jitter, to give an idea of the density of the points
- On page 8, the authors report a list of attributes which is not consistent with the one reported at the bottom of page 5. I get that date_timestamp has been splitted into hour and day, but then what about the month? Also, the attribute is_holiday has disappeared, but then it is back in Table 2.
- On page 8, I do not agree with the authors on "the only temporal features that do not have regular periodicity are the months and the years". Specifically: months do have a periodicity (seasons).
- In Table 2 and Figure 10 year sin/cos do appear: is it the trigonometric transformation of the number of days since 1st January?

Validity of the findings

The authors have addressed my concerns regarding the presentation of their findings.

Additional comments

The paper has improved; I believe it still requires a little work from the authors to address some minor concerns.

---

## Round 0.3 · accepted · Accept

The paper now satisfies the concerns of the reviewers and can be accepted for publication.